# Threat2Traffic: Multi-Agent Environment Synthesis for Malware Traffic Generation from Threat Intelligence

Haoyang Chen [1 2]   Chang Liu [1 2]   Zhong Guan [1 2]   Junzheng Shi [1 2]   Gaopeng Gou [1 2]   Gang Xiong [1 2]

## Abstract

Data-driven cybersecurity research is fundamentally constrained by the scarcity of labeled datasets, yet acquiring authentic, large-scale malware traffic remains bottlenecked by obsolescent public datasets, unscalable manual construction, and inflexible sandboxes that fail to satisfy the sample-specific dependencies required for malware to exhibit malicious behavior. Threat intelligence documents these dependencies, and LLM agents offer a path to extract them for environment construction, yet directly applying such agents faces two challenges: input-side ambiguity and output-side fragility. In this paper, we propose Threat2Traffic, a multi-agent framework that extracts sample-specific dependencies from threat intelligence, reconstructs tailored environments, and captures malware traffic. To address input-side ambiguity, it formulates dependency extraction as structured multi-agent deliberation over an evidence graph. To overcome output-side fragility, it incorporates invariant-guided synthesis with dual-layer validation under syntactic and semantic constraints. Evaluated on 1,077 samples across eight malware families, Threat2Traffic achieves 83.1% reproduction success, highlighting its effectiveness for scalable and realistic malware traffic generation. We release the core source code and traffic dataset at https://github.com/apos3637/Threat2Traffic.

## 1. Introduction

Training deep learning models in cybersecurity demands large-scale labeled traffic datasets, yet publicly available benchmarks (Tavallaee et al., 2009; Sharafaldin et al., 2018; Moustafa & Slay, 2015) are now 5–25 years old (Appendix A), while manual environment construction remains prohibitively costly. Automated methods rely on generic pre-configured sandboxes that cannot adapt to sample-specific dependencies. This creates a critical gap: modern malware increasingly employs environment-aware evasion techniques (Vouvoutsis et al., 2025; Afianian et al., 2019), verifying specific dependencies before exhibiting malicious behavior. With over 80% of samples exhibiting such behaviors (Galloro et al., 2022; Kondracki et al., 2022), generic sandboxes often fail these checks, causing malware to remain dormant.

Triggering malware behavior thus requires constructing environments that satisfy each sample's specific dependencies (Vouvoutsis et al., 2025; Miramirkhani et al., 2017; Alaeiyan et al., 2019). Threat intelligence documents these dependencies, but such information is scattered across verbose, heterogeneous sources, and extracting actionable specifications demands sophisticated reasoning. An ideal solution would automate the pipeline from parsing threat intelligence to capturing authentic traffic (Figure 1(a)). While environment provisioning and traffic capture are well-supported by existing tools, the upstream reasoning tasks remain the critical bottleneck: extracting environmental specifications from threat intelligence (Cuong Nguyen et al., 2025; Büchel et al., 2025) and synthesizing Infrastructure-as-Code (IaC) configurations to satisfy sample-specific dependencies (Kon et al., 2024; Xu et al., 2024).

These bottlenecks demand capabilities that align naturally with Large Language Models: comprehending technical documentation (Chen et al., 2021; Roziere et al., 2023), reasoning over implicit constraints, and reconciling heterogeneous sources. Multi-Agent System (MAS) frameworks further augment LLMs with tool use and collaborative reasoning (Yao et al., 2022; Wu et al., 2024; Hong et al., 2023; Guo et al., 2024). However, directly applying these techniques reveals a fundamental *probabilistic-deterministic gap*:

- **Input-side ambiguity**: Environmental constraints are implicit and scattered across verbose documents, compounding "lost-in-the-middle" effects (Huang et al., 2025).

[1]Institute of Information Engineering, Chinese Academy of Sciences, Beijing, China [2]School of Cyber Security, University of Chinese Academy of Sciences, Beijing, China. Correspondence to: Chang Liu <liuchang@iie.ac.cn>.

Moreover, distinct threat intelligence sources often yield contradictory constraints that cannot be resolved through simple aggregation.

- **Output-side fragility**: IaC languages enforce strict schemas where a single hallucinated field renders the entire configuration non-executable (Zhang et al., 2025; Spracklen et al., 2025).

In this paper, we propose Threat2Traffic, the first multi-agent framework that automatically reasons over threat intelligence to synthesize tailored execution environments and capture authentic malware traffic at scale. Our approach addresses the two challenges through a structured analysis-synthesis decomposition (Figure 1(a)). To resolve input-side ambiguity, we formulate dependency extraction as multi-agent deliberation over an Evidence Graph, where agents propose competing hypotheses from their respective threat intelligence and conflicts are resolved through adversarial arbitration grounded in argumentation theory. To overcome output-side fragility, we introduce an invariant-guided synthesis procedure with dual-layer validation that sequentially enforces syntactic parseability and semantic compliance through iterative refinement. Evaluated on 1,077 samples across 8 malware families, Threat2Traffic achieves 83.1% success, outperforming sandboxes by 17.9% and vanilla LLM baselines by 23%.

The main contributions of this paper are:

- We propose Threat2Traffic, a multi-agent framework that automates malware traffic generation by reasoning over threat intelligence to synthesize sample-specific execution environments.
- To address input-side ambiguity, we introduce a deliberation mechanism (*Dialectic Intent Arbitration*) that constructs an Evidence Graph where agents propose hypotheses and resolve conflicts through adversarial arbitration.
- To address output-side fragility, we develop a constraint-guided synthesis procedure (*Invariant-Guided Synthesis*) with dual-layer validation that enforces syntactic and semantic constraints through iterative refinement.
- We evaluate on 1,077 samples across 8 malware families, achieving 83.1% success rate and outperforming existing baselines by up to 40%. We release the source code and traffic dataset to support future research.

## 2. Problem Formulation

We formulate malware environment synthesis as a **two-stage decomposition** that separates *constraint inference* from *code generation*. This separation addresses input-side ambiguity and output-side fragility respectively.

Let $\mathcal{X}$ denote the space of threat intelligence and $\mathcal{L}$ the formal language of IaC. Our goal is to find:

$$c^* = \arg\max_{c \in \mathcal{M}_\Phi} P(c \mid x) \qquad (1)$$

where $\mathcal{M}_\Phi \subset \mathcal{L}$ is the valid configuration space and $P(c \mid x)$ is approximated via LLM-based generation. A configuration $c \in \mathcal{M}_\Phi$ must satisfy:

- **Syntactic constraints** $\Gamma$: structural rules ensuring parseability of IaC configurations.
- **Semantic constraints** $\Phi$: requirements on configuration *meaning*, comprising *platform-semantic* $\Phi_{\text{plat}}$ (provider schemas) and *sample-semantic* $\Phi_{\text{sample}}(x)$ (environmental specification implicit in $x$).

Inferring $\Phi_{\text{sample}}(x)$ is the core challenge of our problem: it inverts attack graph analysis, recovering configurations from behavioral manifestations rather than deriving behaviors from configurations. This inversion introduces evidential conflict, as distinct threat intelligence sources yield contradictory constraints requiring arbitration, not aggregation.

To bridge this gap, we introduce the *Evidence Graph $\mathcal{G}$* as an intermediate representation (details are provided in Definition 3.1) that makes $\Phi_{\text{sample}}(x)$ explicit before code generation:

$$P(c \mid x) \approx \max_{\mathcal{G}} \Big( P(\mathcal{G} \mid x) \cdot P(c \mid \mathcal{G}, \Gamma, \Phi_{\text{plat}}) \Big) \quad (2)$$

The term $P(\mathcal{G} \mid x)$ corresponds to Stage I, which resolves input-side ambiguity by constructing $\mathcal{G}^*$ through multi-agent deliberation; the term $P(c \mid \mathcal{G}, \Gamma, \Phi_{\text{plat}})$ corresponds to Stage II, which addresses output-side fragility by generating $c$ against known schemas $\Gamma$ and $\Phi_{\text{plat}}$.

## 3. Methodology

Threat2Traffic takes as input a malware sample along with its associated threat intelligence including static analysis results (PE headers, import tables, embedded strings), behavioral execution logs, and external intelligence, and outputs captured network traffic that reflects the malware's authentic behavior. The pipeline proceeds through three phases: (1) extracting structured environmental specifications from threat intelligence, (2) synthesizing executable Infrastructure-as-Code (IaC) from these specifications, and (3) provisioning the environment, executing the sample, and capturing traffic. While the third phase is well-supported by existing toolchains, the first two phases need to address the two challenges identified in Section 1: specification extraction must resolve *input-side ambiguity* from conflicting threat intelligence, and IaC synthesis must overcome *output-side fragility* imposed by rigid provider schemas.

Following the decomposition in Equation 2, Threat2Traffic addresses these challenges through a two-stage framework

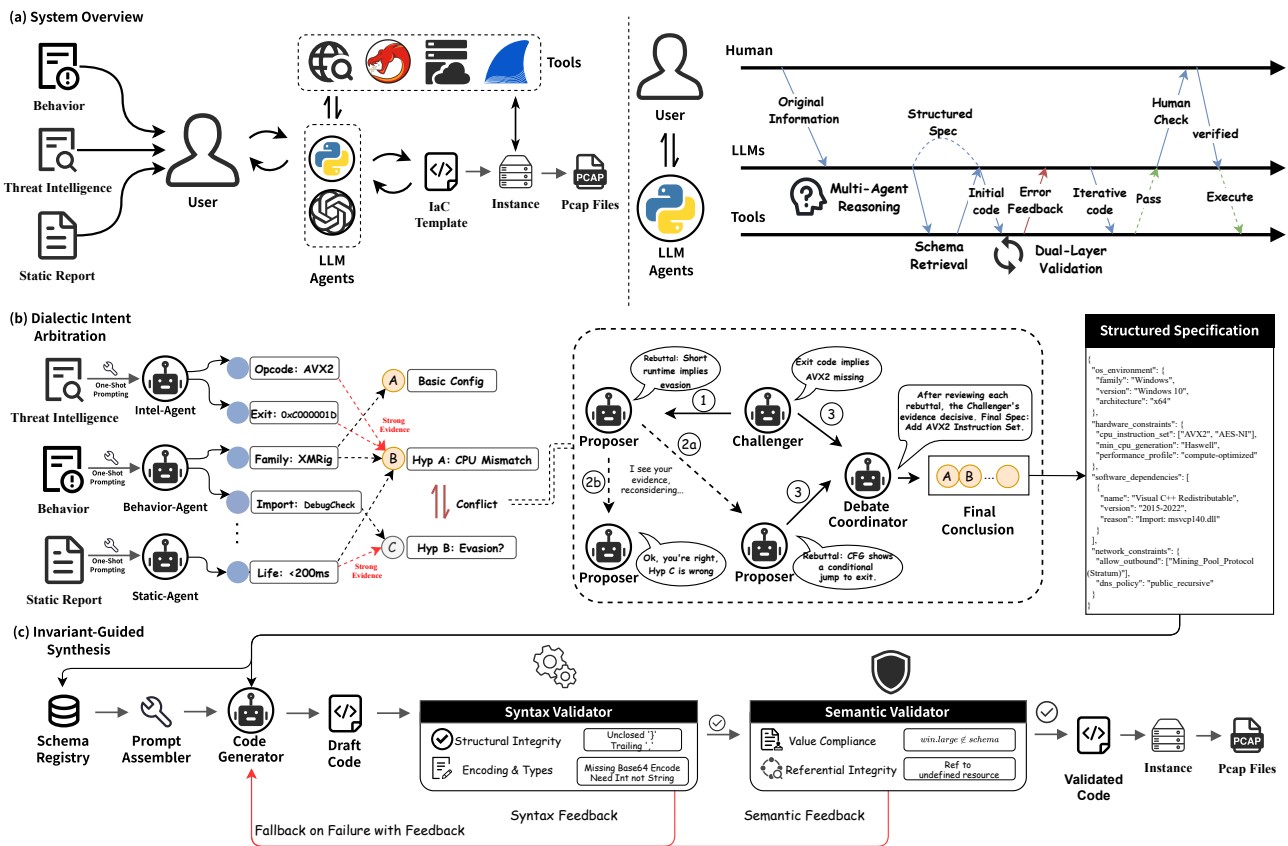

*Figure 1.* **Threat2Traffic Architecture.** (a) End-to-end workflow with multi-agent collaboration. (b) Dialectic Intent Arbitration: agents extract observations into an Evidence Graph and resolve conflicts via adversarial deliberation. (c) Invariant-Guided Synthesis: dual-layer validation ensures executable IaC.

(Figure 1(a)). *Dialectic Intent Arbitration* (Section 3.1) resolves input-side ambiguity by constructing an Evidence Graph through multi-agent deliberation. *Invariant-Guided Synthesis* (Section 3.2) addresses output-side fragility through dual-layer validation against syntactic and semantic constraints. Technical details and worked examples for each stage are provided in Appendix B.

### 3.1. Dialectic Intent Arbitration

This stage produces a structured specification from the threat intelligence, which contains OS requirements, software dependencies, network constraints, and hardware profiles. We structure this process around an Evidence Graph representation (Figure 1(b)).

**Definition 3.1** (Evidence Graph)**.** An Evidence Graph $\mathcal{G} = (\mathcal{V}, \mathcal{E})$ is a directed graph where $\mathcal{V}$ comprises four node types: *observation* (immutable facts from raw data), *hypothesis* (inferred constraints), *reasoning* (deliberation), and *conclusion* (final accepted specifications). Edges $\mathcal{E}$ encode five relationship types that govern information flow:

- `supports` (Observation → Hypothesis): evidential

backing from raw data.
- `debate` (Hypothesis → Hypothesis): directed attack in the argumentation framework, where direction follows the preference function $\pi(h)$.
- `input` (Hypothesis → Reasoning): identifies conflicting hypotheses entering deliberation.
- `modification` (Reasoning → Hypothesis): deliberation produces a corrected hypothesis.
- `validation` (Hypothesis → Conclusion): acceptance into the final specification after grounded extension computation.

**Evidence Graph Construction.** To approximate $P(\mathcal{G} \mid x)$, we deploy three specialized LLM agents: a *Static Analysis Agent*, a *Behavior Analysis Agent*, and a *Threat Intelligence Agent*, each prompted via one-shot prompting to extract observations and hypotheses from their respective sources (detailed prompts are provided in Appendix C.4).

**Construction Invariants.** To ensure well-formed graph structure, the Evidence Graph maintains the following invariants: (1) *Observation Immutability*: observation nodes

**Algorithm 1** Grounded Extension via Fixed-Point Iteration

---

**Require:** Hypothesis set $\mathcal{A}$, Attack relation $\mathcal{R}$
**Ensure:** Grounded extension $E_{gr}$
1: $E_{gr} \leftarrow \emptyset; Defeated \leftarrow \emptyset$
2: **repeat**
3:    $E_{prev} \leftarrow E_{gr}$
4:    $Unattacked \leftarrow \{h \in \mathcal{A} \setminus E_{gr} \mid \nexists h' \in \mathcal{A} \setminus Defeated : (h', h) \in \mathcal{R}\}$
5:    $Defended \leftarrow \{h \in \mathcal{A} \setminus E_{gr} \mid \forall (h', h) \in \mathcal{R} : h' \in Defeated\}$
6:    $E_{gr} \leftarrow E_{gr} \cup Unattacked \cup Defended$
7:    $Defeated \leftarrow \{h \in \mathcal{A} \mid \exists h' \in E_{gr} : (h', h) \in \mathcal{R}\}$
8: **until** $E_{gr} = E_{prev}$
9: **return** $E_{gr}$

---

have no incoming edges and cannot be modified after creation; (2) *Hypothesis Grounding*: every hypothesis must have at least one incoming `supports` edge from an observation, or one incoming `modification` edge from a reasoning node; (3) *Reasoning Completeness*: every reasoning node must have at least two incoming `input` edges and exactly one outgoing `modification` edge; (4) *Conclusion Triggering*: conclusion nodes are created only after the grounded extension $E_{gr}$ is computed, with each $h \in E_{gr}$ producing exactly one `validation` edge.

**Adversarial Deliberation via Argumentation.** As agents initiate the evidence graph from proposed observations and hypotheses, conflicts may emerge. Such conflicts cannot be resolved through naive aggregation. We adapt Dung's Abstract Argumentation Framework (AAF) with three extensions: (1) arguments correspond to *evidence-grounded hypotheses* rather than abstract claims; (2) a preference function $\pi(h)$ directs attacks based on confidence and corroborating observations, enabling asymmetric resolution; and (3) the resulting grounded extension guarantees a *conflict-free*, *well-defended* hypothesis set.

The deliberation proceeds in two steps. *Conflict detection* employs ensemble sampling: we query the LLM $k$ times per hypothesis pair and register a conflict when confidence exceeds threshold $\tau$. *Attack direction* is determined by $\pi(h) = \alpha \cdot \text{CONF}(h) + \beta \cdot \text{SUPPORT}(h)$; an attack edge $(h_i, h_j)$ is added when $h_i$ and $h_j$ conflict and $\pi(h_i) > \pi(h_j)$. Ties ($\pi(h_i) = \pi(h_j)$) are handled conservatively. Details are provided in Appendix B.3.

**Grounded Extension Computation.** With the attack relation established, we compute the grounded extension $E_{gr}$, the unique maximal conflict-free and self-defending hypothesis set, via fixed-point iteration. The accepted hypotheses are serialized into the structured specification (OS requirements, software dependencies, network constraints, and

hardware profiles), providing Invariant-Guided Synthesis with an explicit, conflict-free constraint set. Algorithm details and a worked example are provided in Algorithm 1 and Appendix B.1.

### 3.2. Invariant-Guided Synthesis

This stage produces IaC configurations from the structured specification output by Dialectic Intent Arbitration. To maximize $P(c \mid \mathcal{G}, \Gamma, \Phi_{\text{plat}})$, we employ dual-layer validation that enforces syntactic correctness before semantic compliance, iteratively refining configurations until they become executable (Figure 1(c)).

**Constraint Compilation.** The full platform schema $\Phi_{\text{plat}}$ is too broad for direct optimization. We first compile a local schema by retrieving only the relevant subsets:

$$\Phi_{\text{plat}}^{\text{local}} = \bigcup_{r \in \text{RESOURCES}(\mathcal{G})} \text{SCHEMA}(r) \tag{3}$$

This transforms open-ended generation into a bounded slot-filling task.

**Dual-Layer Validation.** With the local schema compiled, we validate generated configurations through two tiers (Algorithm 2):

- **Tier 1** ($V_\Gamma$): Ensures parseability against IaC grammar $\Gamma$ (e.g., spurious markdown fences, unclosed braces).
- **Tier 2** ($V_\Phi$): Verifies semantic compliance against $\Phi_{\text{plat}}^{\text{local}}$ (e.g., undefined resource references, deprecated fields).

**Algorithm 2** Invariant-Guided Synthesis Loop

---

**Require:** Evidence Graph $\mathcal{G}$, Grammar $\Gamma$, Platform Schema $\Phi_{\text{plat}}$, Max retries $K$
**Ensure:** Valid configuration $c$
1: $\Phi_{\text{plat}}^{\text{local}} \leftarrow \text{COMPILECONSTRAINTS}(\mathcal{G}, \Phi_{\text{plat}})$
2: $c_0 \leftarrow \text{GENERATE}(\mathcal{G}, \Phi_{\text{plat}}^{\text{local}})$
3: **for** $t = 0, \ldots, K$ **do**
4:    $(valid, \mathcal{F}_\Gamma) \leftarrow V_\Gamma(c_t)$
5:    **if** $\neg valid$ **then**
6:       $c_{t+1} \leftarrow \text{REFINE}(c_t, \mathcal{F}_\Gamma)$; **continue**
7:    **end if**
8:    $(valid, \mathcal{F}_\Phi) \leftarrow V_\Phi(c_t, \Phi_{\text{plat}}^{\text{local}})$
9:    **if** $\neg valid$ **then**
10:      $c_{t+1} \leftarrow \text{REFINE}(c_t, \mathcal{F}_\Phi)$; **continue**
11:    **end if**
12:    **return** $c_t$
13: **end for**
14: **return** FAILURE

---

**Iterative Refinement.** When validation fails, we refine the configuration via structured error feedback. The feedback signal $\mathcal{F}$ comprises error location, error type, and remediation hint derived from $\Phi_{\text{plat}}^{\text{local}}$:

$$c_{t+1} \leftarrow \text{REFINE}_{\text{LLM}}(c_t, \mathcal{F}_t) \qquad (4)$$

This loop repeats until validation passes or maximum iterations $K$ is reached, yielding an executable configuration for environment provisioning.

A worked example illustrating the three-step pipeline is provided in Appendix B.2. When Algorithm 2 returns FAILURE, the corresponding samples are considered IaC generation failures. Details about failure analysis are provided in Section 5.

## 4. Results and Discussion

We conduct experiments to evaluate Threat2Traffic's effectiveness in environment reproduction, the contributions of individual components, and practical scalability.

### 4.1. Experiment Setup

**Dataset.** We construct a diverse evaluation dataset comprising 1,077 real-world malware samples across 8 categories: RAT (188), Ransomware (159), Downloader (85), Vidar (89), Infostealer (156), Coinminer (200), Spyware (100), and Adware (100). Samples are sourced from MalwareBazaar (abuse.ch, 2020). File formats are predominantly PE32/PE32+ executables (654 samples, 60.7%), text/scripts (379 samples, 35.2%) and other file types (44 samples, 4.1%). Details on threat intelligence and dataset annotation are provided in Appendix C.1.

**Ground Truth Construction.** To ensure evaluation independence, we employed a *source-separated protocol* that partitions data sources between system inputs and ground truth. System inputs derive from static analysis, threat intelligence and behavioral reports from CAPE (either via VirusTotal or local execution), sources that reveal the malware's potential capabilities. Ground truth behaviors are extracted from other sandboxes within VirusTotal (Zenbox, Yomi Hunter, etc.) whose outputs are never exposed to our system. Two security experts validated behavioral profiles, with inter-annotator agreement reaching Cohen's $\kappa = 0.82$. Details are provided in Appendix C.2.

**Baselines.** We compare against four baselines spanning two categories. *Static sandboxes* include **CAPE** (O'Reilly & Brukhovetskyy) (open-source) and **Hybrid Analysis** (CrowdStrike, 2024) (cloud-based). *LLM-based methods* include **Single LLM** using DeepSeek-Chat (Liu et al., 2025) with up to 8 rounds of human-guided error correction, and

**Vanilla MAS** with majority voting aggregation (Wu et al., 2024). Details about baseline configuration are provided in Appendix C.

**Metrics.** We utilize three metrics. *Infrastructure Provisioning Success Rate (IPSR)* measures the proportion of samples for which the generated IaC passes both syntax and semantic validation. *Malware Behavior Triggering Rate (MBTR)* measures the proportion of ground-truth behaviors successfully triggered, defined as $\text{MBTR}(s) = |B_s \cap \hat{B}_s|/|B_s|$. Here, $B_s$ denotes behavioral profile from ground-truth. *Environment Reproduction Success Rate (ERSR)* is the composite end-to-end metric, computed as $\text{ERSR} = \text{IPSR} \times \text{MBTR}$. Note that for static sandboxes, IPSR is effectively 100% since they use pre-configured environments; thus, ERSR reduces to MBTR.

**Implementation.** All experiments were conducted on a server equipped with an AMD EPYC 7Y83 processor, 300GB RAM, and two NVIDIA A100-40GB GPUs, running Python 3.13. To ensure consistent malware execution regardless of external C2 availability, all synthesized environments include INetSim as a network simulation layer. INetSim allows malware to establish connections and exhibit network behaviors even when original C2 infrastructure is offline, but the connection has no meaningful payload.

**Hyperparameters.** For Dialectic Intent Arbitration, we set ensemble sampling count $k = 5$, conflict threshold $\tau = 0.7$, and preference weights $\alpha = 0.7$, $\beta = 0.3$. For Invariant-Guided Synthesis, we set maximum refinement iterations $K = 8$. Sensitivity analysis confirms robustness across hyperparameter variations, details are provided in Appendix D.3.

### 4.2. Results on Environment Reproduction

We evaluate environment reproduction performance of selected baselines and Threat2Traffic with two backbone LLMs: Qwen3-14B and Claude-Sonnet-4.5. All LLM-based experiments are conducted over 3 independent runs to ensure generalizability; statistical significance analysis is provided in Appendix D.1. Results on environment reproduction are detailed in Table 1.

Threat2Traffic with Claude-Sonnet-4.5 achieves **83.1%** ERSR, outperforming the best sandbox by 17.9% and the best LLM baseline by 23.2%. The framework attains both high MBTR (86.2%) and high IPSR (96.3%). The former reflects adversarial deliberation's ability to extract sample-specific constraints that static sandboxes cannot accommodate; the latter demonstrates dual-layer validation's effectiveness in enforcing schema compliance that unstructured generation frequently violates.

*Table 1.* **End-to-End Performance Comparison.** Performance across baselines and Threat2Traffic with different backbone LLMs. **Columns:** I = IPSR (%), M = MBTR (%), E = ERSR (%). "-" indicates IPSR is 100% by definition for pre-configured environments.

| | Sandbox | | | | | | LLM-based Method | | | | | | Threat2Traffic (Ours) | | | | | |
|---|---|---|---|---|---|---|---|---|---|---|---|---|---|---|---|---|---|---|
| | CAPE | | | Hybrid | | | Single LLM | | | Vanilla MAS | | | Qwen3-14B | | | Claude-Sonnet-4.5 | | |
| **Family** | I | M | E | I | M | E | I | M | E | I | M | E | I | M | E | I | M | E |
| RAT | - | 38.3 | 38.3 | - | 70.2 | 70.2 | 58.4 | 77.2 | 45.1 | 73.5 | 76.8 | 56.4 | 90.2 | 68.3 | 61.6 | 95.2 | 83.7 | **79.7** |
| Ransomware | - | 23.5 | 23.5 | - | 58.5 | 58.5 | 62.1 | 67.4 | 41.8 | 71.2 | 74.5 | 53.0 | 89.5 | 65.4 | 58.6 | 94.8 | 79.3 | **75.2** |
| Downloader | - | 37.7 | 37.7 | - | 60.0 | 60.0 | 65.3 | 73.0 | 47.7 | 76.8 | 78.2 | 60.1 | 91.8 | 69.7 | 64.0 | 96.1 | 83.2 | **80.0** |
| Vidar | - | 70.8 | 70.8 | - | 73.0 | 73.0 | 72.5 | 81.5 | 59.1 | 79.4 | 82.3 | 65.3 | 92.4 | 74.1 | 68.5 | 97.3 | 88.5 | **86.1** |
| Infostealer | - | 57.1 | 57.1 | - | 56.3 | 56.3 | 55.8 | 80.0 | 44.6 | 72.6 | 77.4 | 56.2 | 90.8 | 69.8 | 63.4 | 95.6 | 82.6 | **79.0** |
| Coinminer | - | 40.5 | 40.5 | - | 61.5 | 61.5 | 68.2 | 75.3 | 51.3 | 78.3 | 80.1 | 62.7 | 93.1 | 77.4 | 72.1 | 98.2 | 95.2 | **93.5** |
| Spyware | - | 32.0 | 32.0 | - | 69.0 | 69.0 | 59.4 | 77.7 | 46.2 | 71.8 | 75.6 | 54.3 | 88.6 | 65.0 | 57.6 | 93.8 | 78.0 | **73.2** |
| Adware | - | 41.0 | 41.0 | - | 73.0 | 73.0 | 74.6 | 84.3 | 62.9 | 82.5 | 86.7 | 71.5 | 94.5 | 89.2 | 84.3 | 99.1 | 99.3 | **98.4** |
| **Avg** | - | 42.6 | 42.6 | - | 65.2 | 65.2 | 64.5 | 77.1 | 49.9±4.2 | 75.8 | 79.0 | 59.9±3.6 | 91.4 | 72.4 | 66.3±3.3 | 96.3 | 86.2 | **83.1**±2.0 |

## 4.3. Component Analysis

We evaluate each stage's contribution through two experiments: (1) extraction quality assessment using F1-score against human-annotated constraints, and (2) ablation study removing Adversarial Deliberation, Syntax Validation $V_\Gamma$, and Semantic Validation $V_\Phi$ respectively. All experiments use Claude-Sonnet-4.5 as backbone.

**Constraint Extraction Quality.** Threat2Traffic achieves an average F1-score of **81.0%** on constraint extraction. Categories with explicit dependencies (Adware: 85.7%) outperform those with implicit constraints (Spyware: 74.6%). Higher extraction quality correlates with fewer refinement iterations in synthesis (Spearman's $\rho = -0.87$, $p < 0.01$); per-category breakdown is provided in Appendix D.2.

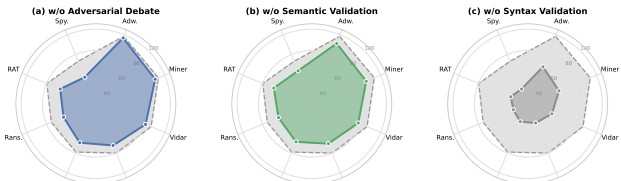

*Figure 2.* **Ablation Study.** ERSR (%) under component removal. (a) w/o Adversarial Deliberation, (b) w/o Semantic Validation, (c) w/o Syntax Validation. Gray dashed lines indicate full approach performance.

**Ablation Study.** We isolate the contribution of each component: Adversarial Deliberation (within Dialectic Intent Arbitration), Syntax Validation $V_\Gamma$, and Semantic Validation $V_\Phi$ (within Invariant-Guided Synthesis). Results are detailed in Figure 2.

*Adversarial Deliberation.* Removing this component causes a **16.2% ERSR drop** for Spyware but only 1.6% for Adware. This disparity reflects constraint explicitness: Spyware's ambiguous signals require principled arbitration,

whereas Adware's explicit dependencies produce consistent outputs where naive voting suffices. This confirms that the value of adversarial deliberation is proportional to input-side ambiguity.

*Dual-Layer Validation.* Removing Syntax Validation causes universal degradation to **52.3% average ERSR**, even for categories with explicit dependencies, demonstrating that $V_\Gamma$ is a non-negotiable prerequisite. In contrast, removing Semantic Validation yields category-dependent degradation: Coinminer drops 12.3% due to strict schema requirements, while RAT shows only 4.2% loss. This confirms $V_\Gamma$ as a universal gatekeeper, with $V_\Phi$ providing schema-specific refinement.

## 4.4. Scalability Analysis

We evaluate scalability across three dimensions: (1) model size using Qwen3 family (0.6B–32B), (2) cross-vendor generalization using 14B-class models (Qwen3, DeepSeek-R1-Distill, Ministral), and (3) frontier model performance (Claude-Sonnet-4.5, GPT-5.1, Gemini-3-flash, DeepSeek-Chat).

**Model Scaling.** The 14B model achieves **66.3%** ERSR, first surpassing Hybrid Analysis (65.2%) and establishing the cost-effective deployment threshold. Models below 4B remain under CAPE (42.6%); 32B provides only marginal gains (+4%) over 14B. Details are provided in Figure 3.

**Cross-Vendor Comparison.** DeepSeek-R1-Distill-14B achieves 67.95%, Qwen3-14B 66.3%, and Ministral-14B 63.66%. The cross-vendor variance (<5%) confirms that performance gains are robust across model providers, suggesting that the structured framework rather than model-specific capabilities drives improvement. Details are provided in Figure 4.

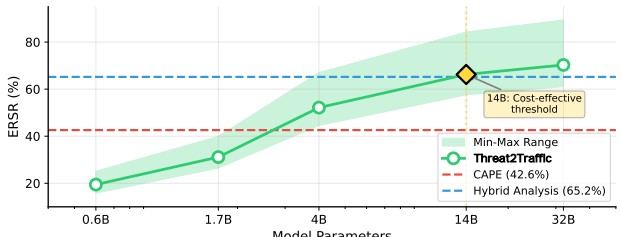

*Figure 3.* **Model Scaling.** ERSR (%) across Qwen3 model sizes (0.6B–32B) with baseline references, evaluating how model capacity affects environment reproduction performance.

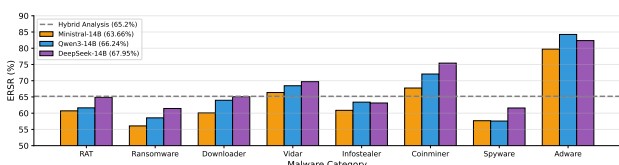

*Figure 4.* **Cross-Vendor 14B Comparison.** ERSR (%) of 14B-class models across malware categories, examining whether performance generalizes across model providers at comparable scale.

**Frontier Model Performance.** Claude-Sonnet-4.5 achieves **83.1%**, with the largest relative gains in high-ambiguity categories (Spyware: +27.1%) compared to explicit-dependency categories (Adware: +16.7%). This pattern confirms that stronger reasoning capabilities primarily benefit adversarial deliberation, as resolving conflicting evidence requires sophisticated inference that frontier models provide. In contrast, constraint-guided synthesis shows more modest gains since schema compliance is largely enforced by structural validation rather than model intelligence. Details are provided in Figure 6.

### 4.5. Practical Utility

High reproduction success rate alone does not guarantee practical value. We evaluate whether Threat2Traffic delivers authentic traffic and operational benefits in real-world deployment through two experiments: (1) traffic authenticity validation, comparing captured protocol distributions against known family-specific behavioral patterns, and (2) operational efficiency comparison, measuring setup time, analysis throughput, and intelligence yield against production sandboxes.

**Traffic Authenticity** We validate whether captured traffic reflects genuine malware behaviors by comparing protocol distributions against known family-specific patterns. The distributions align with expected operational patterns: Adware predominantly uses HTTP for ad injection, while Infostealers favor HTTPS for credential exfiltration. These category-specific signatures confirm that synthesized environments trigger authentic execution rather than producing

generic simulation noise. Details are provided in Figure 5.

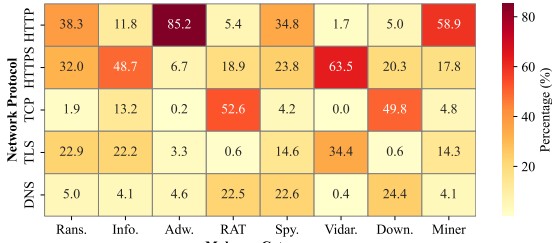

*Figure 5.* **Traffic Authenticity Validation.** Protocol distribution (%) across 8 malware categories, verifying whether triggered behaviors match known family-specific network patterns.

**Operational Efficiency** We compare operational efficiency and intelligence utility against CAPE and Hybrid Analysis. With comparable setup time to Hybrid Analysis (420–600s vs 120–600s), Threat2Traffic activates **895 valid samples** versus 702, a 27.5% improvement. This gain stems from adaptive synthesis: by satisfying sample-specific constraints, Threat2Traffic triggers behaviors in samples that remain dormant in static sandboxes. The resulting intelligence yield improves accordingly, with 18% more IoCs per sample (12.6 vs 10.7) and 113 unique MITRE TTPs versus 105. Details are provided in Table 2.

*Table 2.* **Operational Efficiency and Intelligence Utility.** Setup time measures environment preparation; analysis time measures per-sample execution. Valid samples indicates successful behavior triggering.

|  | CAPE | Hybrid Analysis | Threat2Traffic |
|---|---|---|---|
| ***Operational Efficiency*** | | | |
| Setup Time | ∼24h | 120–600s | ∼420s–600s[*] |
| Avg. Analysis Time | 300s | >300s | ∼**157s**[*] |
| ***Intelligence Utility*** | | | |
| Valid Samples | 459 | 702 | **895** |
| Avg. IoCs / Sample | 8.4 | 10.7 | **12.6** |
| Unique TTPs | 82 | 105 | **113** |

[*]Varies with instance configuration.

## 5. Limitations and Future Work

**Failure Limitations.** Despite achieving 83.1% success, understanding failure modes is essential for future improvements. We analyzed 182 failed samples to identify systematic limitations. Figure 7 summarizes the failure distribution:

- **Synthesis and Inference Failures (42.3%).** IaC generation failures (22.0%) arise from unsupported architectures or deprecated OS images unavailable from cloud providers. Uninferable constraints (20.3%) include geofencing, browser extension dependencies, and software checks beyond available threat intelligence. Enhanced

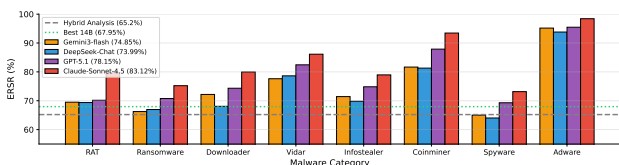

*Figure 6.* **Frontier Model Comparison.** ERSR (%) of frontier LLMs across malware categories, assessing performance upper bounds with the most capable models.

constraint inference through binary instrumentation or symbolic execution represents a promising direction.

- **Infrastructure and Interaction Barriers (51.7%).** Sample inactivation (28.6%) occurs when malware requires *semantically valid* C2 responses beyond generic simulation. While INetSim provides protocol-level emulation for common services, certain samples perform cryptographic handshakes, expect specific command structures, or validate certificates, which simulation cannot satisfy. This represents a fundamental ceiling requiring either live C2 infrastructure or family-specific response emulation.
- **Other Causes (6.0%).** The remaining failures comprise edge cases including corrupted samples and sophisticated anti-analysis techniques.

**Ground Truth Limitations.** Beyond system failures, our evaluation methodology has inherent constraints. We rely on behavioral profiles extracted from alternative sandboxes as ground truth. However, these sandboxes may miss certain behaviors due to their environmental limitations, potentially causing our reported ERSR to underestimate the true reproduction rate. Additionally, the source-separated protocol, while preventing data leakage, means that ground truth behaviors may include artifacts specific to those sandbox environments that Threat2Traffic's synthesized environments cannot replicate by design.

**Future Work.** We identify two primary directions for future research. First, *targeted user interaction simulation*: currently, sample execution after environment construction relies on shell scripts; since 23.1% of failures stem from additional interaction requirements (e.g., specific website visits, credential entry, form submissions), developing agents capable of goal-directed, context-aware user behavior emulation would significantly expand coverage. Second, *cross-platform extension*: our current implementation focuses on Windows and Linux malware; extending Threat2Traffic to mobile (Android/iOS) and IoT platforms would broaden its applicability, though each platform presents unique challenges in environment specification and behavioral profiling.

# 6. Related Work

**Malware Analysis Environments.** Dynamic analysis systems (O'Reilly & Brukhovetskyy; CrowdStrike, 2024) automate malware execution but lack adaptability to anti-analysis techniques (Afianian et al., 2019; Miramirkhani et al., 2017). Bare-metal execution (Kirat et al., 2014) counters some evasion but remains prohibitively expensive. Neither approach addresses the need for sample-specific environment synthesis.

**LLMs for Threat Intelligence and Code Generation.** Large language models offer new capabilities for both information extraction and code generation. Recent work has applied LLMs to extract structured information from threat intelligence, including attack patterns (Li et al., 2022; Satvat et al., 2021), indicators of compromise (Gao et al., 2023), and MITRE ATT&CK mappings (Büchel et al., 2025). However, these approaches reason *forward* from threat reports to attack techniques, whereas environment synthesis requires the *inverse*. This inversion introduces evidential ambiguity absent in forward extraction, as multiple configurations may produce identical behaviors. Separately, LLM-based code generation has shown promise for Infrastructure-as-Code (Kon et al., 2024; Xu et al., 2024), yet schema brittleness causes frequent failures without structured validation (Rahman et al., 2019). Our work bridges these two lines by combining multi-agent deliberation for constraint inference with dual-layer validation for reliable IaC synthesis.

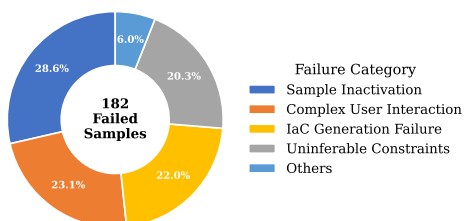

*Figure 7.* **Failure Case Analysis.** Distribution of failure reasons across 182 unsuccessful samples, comprising synthesis failures and execution failures.

**Multi-Agent Deliberation and Constrained Generation.** Multi-agent debate improves LLM reasoning (Du et al., 2023; Liang et al., 2024), but general debate lacks principled conflict resolution. We adapt Dung's Abstract Argumentation Framework (Dung, 1995) with preference-based extensions (Modgil, 2009) for evidence-weighted arbitration. For IaC generation, schema brittleness causes frequent failures; even frontier LLMs achieve under 30% success without feedback (Kon et al., 2024; Xu et al., 2024; Rahman et al., 2019). Existing remedies such as constrained

decoding (Geng et al., 2023; Willard & Louf, 2023; Park et al., 2024; Mündler et al., 2025), RAG (Lewis et al., 2020; Ayala & Bechard, 2024), and self-refinement (Madaan et al., 2023) address syntax but neglect semantics; our dual-layer validation enforces both.

## 7. Conclusion

We presented Threat2Traffic, the first multi-agent framework addressing malware environment synthesis as an *inverse* of attack graph analysis, recovering enabling configurations from observed behaviors. Our approach separates the problem into two stages: *Dialectic Intent Arbitration* produces conflict-free hypothesis sets via grounded extension computation, while *Invariant-Guided Synthesis* achieves 96.3% provisioning success by reducing LLM hallucinations through syntactic and semantic validation. Together, these stages enable the complete pipeline from threat intelligence to authentic traffic capture.

Our experiments validate that this two-stage decomposition effectively bridges the probabilistic-deterministic gap: the Evidence Graph with adversarial deliberation resolves input-side ambiguity, while dual-layer validation eliminates output-side fragility. Evaluated on 1,077 real-world malware samples, Threat2Traffic achieves 83.1% reproduction success rate, significantly outperforming existing sandbox solutions and addressing the data scarcity bottleneck in cybersecurity research.

## Acknowledgments

We thank the anonymous reviewers for their valuable feedback. We are also grateful to our colleagues for the insightful discussions throughout this work. We also thank `abuse.ch` and the VirusTotal community for maintaining the public threat intelligence resources that made this research possible. This work was supported by the National Natural Science Foundation of China under Grant No. 62402492.

## Impact Statement

**Broader Impact.** This work aims to address the critical data scarcity problem in cybersecurity research by enabling scalable generation of high-fidelity malware network traffic datasets. The availability of labeled datasets is essential for developing systems that protect individuals, organizations, and critical infrastructure from cyber threats. By automating the environment reproduction process, our framework can accelerate security research and reduce the manual effort required from expert analysts.

**Potential Risks and Mitigations.** We acknowledge that tools designed for malware analysis inherently carry dual-use concerns. While Threat2Traffic is intended to support defensive security research, the generated Infrastructure-as-Code templates could theoretically be misused to facilitate malware testing by malicious actors. We implement several safeguards to mitigate these risks:

- All malware samples used in this study are sourced exclusively from MalwareBazaar (abuse.ch, 2020), with threat intelligence metadata retrieved from VirusTotal (Google, 2024).
- The dataset contains only network traffic traces, not executable malware binaries.
- The released source code focuses on threat intelligence analysis and code generation, excluding components that could facilitate malware distribution or evasion.
- The framework is designed for controlled research environments and includes no capabilities for active network attacks or payload delivery.

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

*Table 3.* **Representative Public Network Traffic Datasets.** We categorize datasets by primary use case and highlight temporal distribution. Datasets marked with † are most frequently cited in recent IDS literature (Tavallaee et al., 2009; Sharafaldin et al., 2018; Moustafa & Slay, 2015; Garcia et al., 2014; 2020).

| Dataset | Year | Scale | Source | Format | Primary Use Case |
|---|---|---|---|---|---|
| *IDS Benchmarks* | | | | | |
| KDD Cup 1999† | 1999 | 4.9M records | UCI/SIGKDD | CSV | IDS benchmark (legacy) |
| NSL-KDD† | 2009 | 148K records | UNB CIC | ARFF/CSV | Improved IDS benchmark |
| UNSW-NB15† | 2015 | 2.5M records | UNSW | PCAP/CSV | Modern attack types |
| CICIDS2017† | 2017 | 2.8M+ flows | UNB CIC | PCAP/CSV | Comprehensive IDS |
| CSE-CIC-IDS2018 | 2018 | 16.2M instances | CSE/CIC | PCAP/CSV | Cloud-scale IDS |
| *Malware & Botnet Traffic* | | | | | |
| CTU-13 | 2014 | 13 scenarios | CTU Prague | PCAP/NetFlow | Botnet detection |
| ISOT Botnet | 2010 | 1.6M packets | U. Victoria | PCAP | Storm/Waledac botnets |
| IoT-23 | 2020 | 325M flows | CTU/Avast | PCAP/Zeek | IoT malware families |
| *IoT Security* | | | | | |
| N-BaIoT | 2018 | 7.0M records | Ben-Gurion U. | CSV | Mirai/BASHLITE detection |
| Bot-IoT | 2019 | 72M+ records | UNSW | PCAP/CSV | IoT forensics |
| CICIoT2023 | 2023 | 46.7M records | UNB CIC | PCAP/CSV | Large-scale IoT |
| *Encrypted Traffic* | | | | | |
| ISCXVPN2016 | 2016 | 28GB | UNB CIC | PCAP/CSV | VPN traffic classification |
| ISCXTor2016 | 2016 | 22GB | UNB CIC | PCAP/CSV | Tor traffic analysis |
| CSTNET-TLS1.3 | 2021 | 120 app classes | CSTNET/CAS | TSV | TLS 1.3 classification |

# A. Survey of Public Network Traffic Datasets

To contextualize the data scarcity challenge addressed by Threat2Traffic, we surveyed over 20 publicly available network traffic datasets commonly used in cybersecurity research. This appendix summarizes the landscape and highlights the temporal and methodological gaps that motivate our work.

### A.1. Dataset Taxonomy

Table 3 presents a representative subset of major public datasets, categorized by primary use case. The complete survey covers intrusion detection, malware traffic analysis, DDoS attacks, botnet detection, IoT security, and encrypted traffic classification.

**Temporal Obsolescence.** Among the 20+ surveyed datasets, approximately 55% were released before 2020. The four most-cited IDS benchmarks (KDD Cup 1999, NSL-KDD, UNSW-NB15, CICIDS2017) have median age exceeding 8 years. This temporal gap is particularly problematic given the rapid evolution of malware: techniques such as living-off-the-land binaries (LOLBins), fileless malware, and sophisticated anti-sandbox checks were rare or nonexistent when these datasets were created.

### A.2. Major Dataset Repositories

For researchers seeking to access public datasets, we list the primary repositories:

- **CIC Datasets Portal**: https://www.unb.ca/cic/datasets/ hosts CICIDS2017, CIC-DDoS2019, CICIoT2023, and related datasets from the Canadian Institute for Cybersecurity.
- **UNSW Research Data**: https://research.unsw.edu.au/ provides UNSW-NB15, Bot-IoT, and TON_IoT datasets.
- **Stratosphere IPS**: https://www.stratosphereips.org/datasets-overview offers CTU-13, IoT-23, and ongoing malware captures.
- **IEEE DataPort**: https://ieee-dataport.org/ aggregates community-contributed security datasets.
- **UCI ML Repository**: https://archive.ics.uci.edu/ hosts KDD Cup 1999 and N-BaIoT.

## B. Methodology Details

This appendix provides technical details for the two-stage methodology presented in Section 3.

### B.1. Grounded Extension Computation

We compute the grounded extension $E_{gr}$, the unique maximal set of defensible arguments, via fixed-point iteration. The algorithm iteratively identifies unattacked hypotheses, defended hypotheses (whose attackers are defeated), and defeated hypotheses until convergence.

**Worked Example.** Consider a malware sample analysis where Stage I agents have extracted the hypotheses shown in Table 4(a). Using ensemble sampling ($k = 5$), we identify conflicting hypothesis pairs: $(h_1, h_2)$ yields conflict confidence $c = 0.80 \geq 0.7$ for .NET Framework versions, and $(h_3, h_4)$ yields $c = 1.00 \geq 0.7$ for OS targets.

The preference function $\pi(h) = 0.7 \cdot \text{CONF}(h) + 0.3 \cdot \text{SUPPORT}(h)$ determines attack direction, as shown in Table 4(b). Since $\pi(h_1) > \pi(h_2)$ and $\pi(h_3) > \pi(h_4)$, the attack relation is $\mathcal{R} = \{h_1 \to h_2, \ h_3 \to h_4\}$.

*Table 4.* Grounded Extension computation example. (a) Extracted hypotheses. (b) Preference scores and attack relations.

**(a) Hypotheses**

| ID | Content | Conf. | Supp. |
|----|---------|-------|-------|
| $h_1$ | .NET Framework 4.5 | High | 3 |
| $h_2$ | .NET Framework 3.5 | Medium | 1 |
| $h_3$ | Target: Windows 10 | High | 4 |
| $h_4$ | Target: Windows 7 | Low | 1 |
| $h_5$ | Admin privileges | High | 2 |

**(b) Preference Scores**

| Hyp. | $\pi(h)$ | Attack |
|------|----------|--------|
| $h_1$ | $0.7 \times 3 + 0.3 \times 3 = 3.0$ | $h_1 \to h_2$ |
| $h_2$ | $0.7 \times 2 + 0.3 \times 1 = 1.7$ | |
| $h_3$ | $0.7 \times 3 + 0.3 \times 4 = 3.3$ | $h_3 \to h_4$ |
| $h_4$ | $0.7 \times 1 + 0.3 \times 1 = 1.0$ | |
| $h_5$ | $0.7 \times 3 + 0.3 \times 2 = 2.7$ | (none) |

In Iteration 1, unattacked hypotheses $h_1$, $h_3$, and $h_5$ are accepted into $E_{gr}$, and $h_2$, $h_4$ are marked as defeated. In Iteration 2, no new hypotheses can be accepted, so the algorithm converges with $E_{gr} = \{h_1, h_3, h_5\}$. The final specification requires .NET Framework 4.5, targets Windows 10, and requires administrator privileges.

### B.2. Stage II Worked Examples

This subsection illustrates the *Invariant-Guided Synthesis* pipeline through a concrete example.

**Step 1: Constraint Compilation.** Given an input specification containing target environment constraints (e.g., Windows Server 2019, x86_64 architecture) and inferred dependencies (e.g., .NET Framework 3.5, VC++ Redistributable), the Resource Inference module determines required cloud resources: `instance`, `vpc`, `subnet`, `user_code`, and `security_group` (Figure 8).

**Step 2: Initial Generation and Dual-Layer Validation.** The Prompt Assembler retrieves schema definitions from the Schema Registry, specifying field constraints such as valid `instance_type` enumerations and `image_id` patterns. The Code Generator produces an initial draft containing both syntactic errors (e.g., hallucinated attributes, malformed heredoc delimiters) and semantic errors (e.g., invalid image IDs, dangling references). The Syntax Validator detects these violations (Figure 9a).

**Step 3: Iterative Refinement.** In the first iteration, the Syntax Validator's feedback guides removal of hallucinated attributes and correction of heredoc formatting. The resulting code passes syntax validation but contains semantic errors. In the second iteration, the Semantic Validator identifies value compliance issues and referential integrity violations. By querying the Schema Registry, the Code Generator produces the final corrected code (Figure 9b).

### B.3. Handling Symmetric Conflicts

When two conflicting hypotheses have equal preference scores ($\pi(h_i) = \pi(h_j)$), our framework registers bidirectional attacks, forming a symmetric cycle. Under Dung's grounded semantics, neither hypothesis can be defended, and both are excluded from the extension.

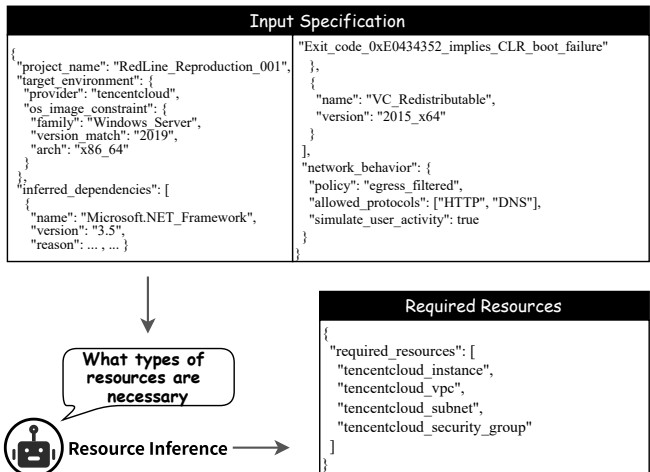

*Figure 8.* **Step 1: Resource Inference.** The system analyzes the input specification to determine required cloud resources.

This conservative behavior aligns with the security principle of favoring precision over recall: when evidence strength is equally ambiguous, the system abstains rather than commits to a potentially incorrect environmental constraint.

**Empirical Analysis.** Across our evaluation dataset, symmetric conflicts occurred in only 2.8% of all detected conflicts (27 out of 963 conflict pairs). The resulting conservative exclusions affected 2.1% of final specifications. Manual inspection revealed that 78% of these cases involved genuinely ambiguous scenarios where human analysts also expressed uncertainty.

**Alternative Strategies.** For applications requiring higher recall, alternative resolution strategies can be employed: (1) *Random tie-breaking*: randomly select one hypothesis to attack the other; (2) *Recency bias*: prefer hypotheses supported by more recent observations; (3) *Human-in-the-loop*: escalate symmetric conflicts to human analysts. We leave these strategies as part of our future work.

## C. Experimental Setup

This appendix provides detailed information on the experimental setup summarized in Section 4.

### C.1. Dataset and Threat Intelligence Sources

Our framework extracts environmental constraints from specific fields that do not require complete behavioral execution. From static analysis, we leverage PE headers, import tables, and embedded strings to infer OS requirements and runtime dependencies. Threat intelligence feeds contribute malware family signatures and MITRE ATT&CK technique mappings (Strom et al., 2018) that encode known environmental prerequisites. Partial execution logs, even from failed sandbox runs, reveal attempted API calls and registry queries that expose implicit environment checks.

Each sample was annotated with four constraint dimensions: (1) *Runtime Dependencies*—requirements for specific libraries, frameworks, or browser versions; (2) *System Constraints*—hardware or OS-level requirements such as CPU instruction sets or registry keys; (3) *Anti-Analysis*—evasion techniques including sandbox detection and timing checks; and (4) *Interaction Required*—need for user actions such as clicking or form submission.

### C.2. Ground Truth Construction

To ensure evaluation integrity, we employ a *source-separated protocol* that strictly separates input sources from ground truth sources (Table 5). The key constraint is: *any data source used as system input cannot serve as ground truth*.

CAPE behavioral reports are used as input to our system's intent extraction pipeline. If we also used CAPE as ground truth, the system could trivially "succeed" by memorizing associations from its input rather than genuinely inferring environmental requirements. By using only Zenbox, Yomi Hunter, C2AE, and Jujubox for ground truth, we ensure the system must generalize beyond its input sources.

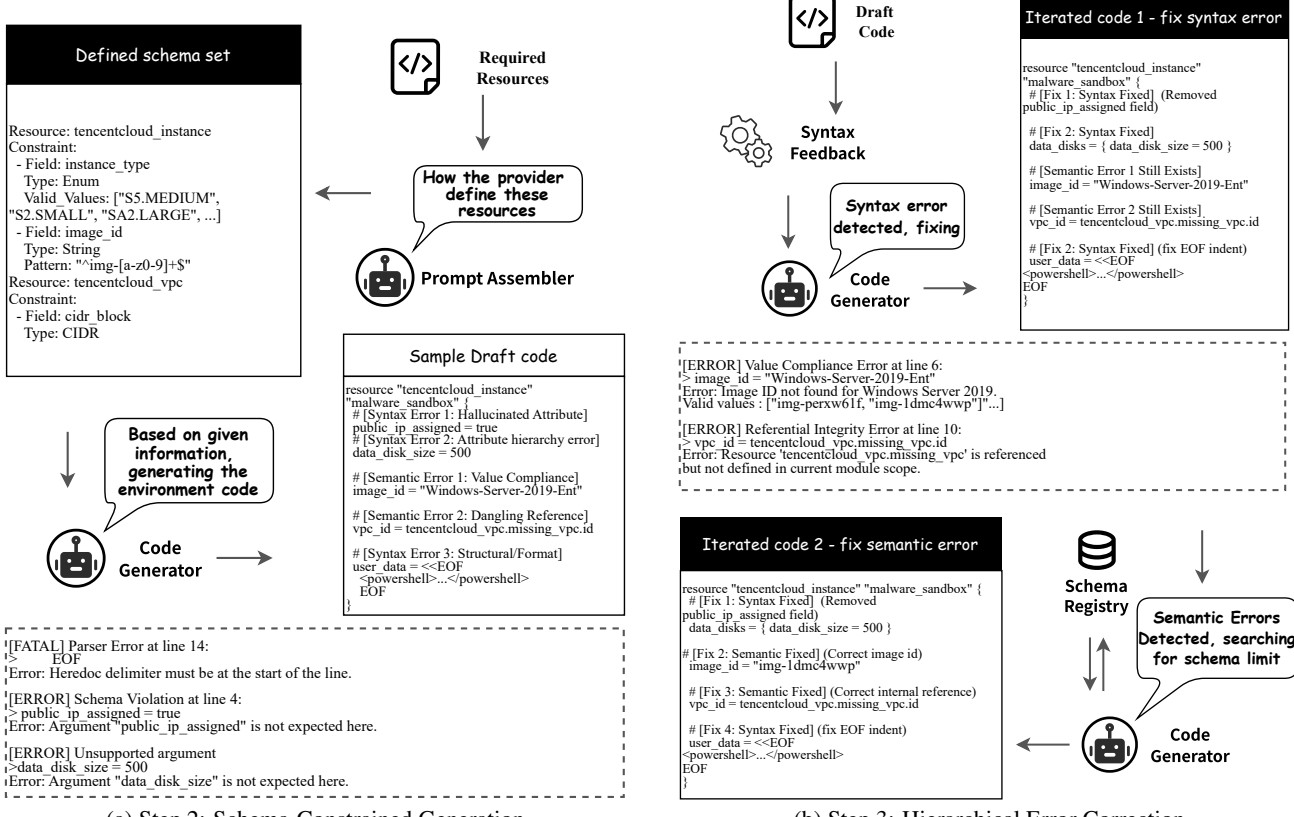

(a) Step 2: Schema-Constrained Generation  (b) Step 3: Hierarchical Error Correction

*Figure 9.* **Invariant-Guided Synthesis validation process.** (a) Initial draft contains syntax and semantic errors detected by validators. (b) Dual-layer validation iteratively fixes syntax then semantic errors.

Two security experts independently validated behavioral profiles against ground truth sandbox reports through: (1) *Behavior Extraction*: extracting observed behaviors (network connections, file operations, registry modifications, process creation); (2) *Capability Mapping*: mapping behaviors to environmental requirements; and (3) *Conflict Resolution*: resolving disagreements via discussion, excluding unresolvable cases. Inter-annotator agreement reached Cohen's $\kappa = 0.82$ (substantial agreement) (Cohen, 1960). Samples without consensus were excluded, yielding the final 1,077 samples.

## C.3. Baseline Configurations

**Single LLM Baseline.** This baseline receives static analysis results, threat intelligence, and behavior logs, then directly generates IaC. Specifically, we provide the model with a comprehensive prompt containing: (a) static analysis results including file metadata, imported libraries, and suspicious API calls; (b) threat intelligence excerpts from public reports; and (c) partial dynamic execution logs. The baseline includes up to 8 rounds of human-guided error correction to isolate our architectural contributions from iterative refinement capability.

**Vanilla MAS Baseline.** This baseline employs a general-purpose multi-agent architecture inspired by AutoGPT. The system consists of three parallel analysis agents: (1) a **Static Analysis Agent** that processes PE headers, import tables, and string artifacts; (2) a **Dynamic Analysis Agent** that analyzes preliminary sandbox execution logs; and (3) a **Threat Intelligence Agent** that queries public threat feeds.

Unlike Threat2Traffic's Evidence Graph with adversarial deliberation, the Vanilla MAS baseline employs a simple aggregation strategy: each agent produces independent recommendations, which are merged by a coordinator agent through majority voting. This approach lacks: (1) structured hypothesis tracking and conflict resolution; (2) grounded extension computation for consistent constraint sets; and (3) iterative refinement through semantic validation feedback.

*Table 5.* Source separation to prevent data leakage. CAPE is excluded from ground truth because its reports are used as system input.

| Family | Sources | Description |
|---|---|---|
| **System Input** | CAPE | Behavioral reports for intent extraction |
| | CAPA | Static capability analysis |
| | VT Detection | Aggregated AV verdicts and family labels |
| | File Metadata | PE headers, imports, strings |
| **Ground Truth** | Zenbox | Intezer's cloud-based behavioral sandbox |
| | Yomi Hunter | Yoroi's malware analysis environment |
| | C2AE | VirusTotal behavioral sandbox |
| | Jujubox | VirusTotal behavioral sandbox |

## C.4. Prompt Templates

### C.4.1. SINGLE LLM BASELINE PROMPT

```
You are a malware analysis expert. Your task is to analyze
the provided artifacts and generate Infrastructure-as-Code
(Terraform) to create an environment capable of executing
the given malware sample.

## Static Analysis Results
File Name: {file_name}
File Type: {file_type}
File Size: {file_size}
MD5: {md5_hash}
SHA256: {sha256_hash}

Imported Libraries:
{imported_libraries}

Suspicious API Calls:
{suspicious_apis}

Strings of Interest:
{interesting_strings}

## Threat Intelligence
{threat_intelligence_excerpt}

## Partial Dynamic Execution Logs
{dynamic_logs}

## Task
Based on the above information, generate a complete
Terraform configuration that provisions a cloud environment
suitable for executing this malware sample. Consider:

1. Operating System: Infer the required OS version from
   the artifacts (e.g., Windows 7/10/11, specific builds)
2. Runtime Dependencies: Identify required frameworks,
   libraries, or software (e.g., .NET, VC++ Runtime,
   specific browser versions)
3. System Configuration: Any registry keys, environment
   variables, or system settings that may be required
4. Network Configuration: Required network access,
   firewall rules, or proxy settings

Output only valid Terraform HCL code without explanation.
```

*Listing 1.* Single LLM Baseline Prompt Template

### C.4.2. VANILLA MAS BASELINE PROMPTS

The Vanilla MAS baseline employs zero-shot prompting without demonstration examples. Each agent independently produces recommendations, which are aggregated by a coordinator through majority voting.

```
You are a malware static analysis expert.
```

```
Analyze the following static artifacts and infer the
environmental requirements for executing this malware.

## Input
{input_data}

## Output
Provide your analysis in the following format:
- Target OS: [your inference]
- Dependencies: [list of required software/libraries]
- Confidence: [high/medium/low]
- Reasoning: [brief explanation]
```

*Listing 2.* Vanilla MAS: Static Analysis Agent

```
You are a malware behavior analysis expert.

Analyze the following behavioral logs and infer the
environmental requirements for executing this malware.

## Input
{input_data}

## Output
Provide your analysis in the following format:
- Target OS: [your inference]
- Dependencies: [list of required software/libraries]
- Confidence: [high/medium/low]
- Reasoning: [brief explanation]
```

*Listing 3.* Vanilla MAS: Behavior Analysis Agent

```
You are a threat intelligence expert.

Analyze the following threat intelligence and infer the
environmental requirements for executing this malware.

## Input
{input_data}

## Output
Provide your analysis in the following format:
- Target OS: [your inference]
- Dependencies: [list of required software/libraries]
- Confidence: [high/medium/low]
- Reasoning: [brief explanation]
```

*Listing 4.* Vanilla MAS: Threat Intelligence Agent

```
You are a coordinator agent. Aggregate the following
analyses from three expert agents using majority voting.

## Agent Analyses
### Static Analysis Agent:
{static_output}

### Behavior Analysis Agent:
{behavior_output}

### Threat Intelligence Agent:
{threat_intel_output}

## Task
For each field (Target OS, Dependencies), select the
most common answer. If there is no majority, choose
the one with highest confidence.

## Output
- Final Target OS: [majority vote result]
- Final Dependencies: [merged list]
- Aggregation Notes: [any conflicts noted]
```

*Listing 5.* Vanilla MAS: Coordinator (Majority Voting)

## C.4.3. THREAT2TRAFFIC SYSTEM PROMPTS

The following prompts employ one-shot prompting to guide Evidence Graph construction. Each prompt includes a demonstration example to establish the expected reasoning pattern and output structure.

```
You are a malware static analysis expert.

## Task
Analyze the provided static information and extract observations
about the malware's environment requirements, dependencies,
and execution constraints.

## Demonstration Example

Input:
- File: WindowsUpdate.exe (PE32+ x86-64, 1.4MB)
- Imports: kernel32.dll (IsDebuggerPresent, CreateProcessW),
           advapi32.dll (RegOpenKeyExA, RegSetValueExA),
           ws2_32.dll (WSASocketW, connect, send, recv)
- Tags: detect-debug-environment, persistence, 64bits

Output:
{
  "observations": [
    {
      "content": "64-bit Windows PE executable (PE32+ x86-64)",
      "category": "os_version",
      "evidence_type": "pe_header",
      "raw_data": {"magic": "PE32+", "arch": "x86-64"}
    },
    {
      "content": "Imports IsDebuggerPresent for anti-debug",
      "category": "anti_analysis",
      "evidence_type": "imports",
      "raw_data": {"dll": "kernel32.dll", "func": "IsDebuggerPresent"}
    }
  ],
  "os_indicators": {
    "target_os": "windows",
    "architecture": "x64",
    "min_version": "Windows 7"
  }
}

## Input Data
{input_data}
```

*Listing 6.* Static Analysis Agent Prompt

```
You are a malware behavior analysis expert.

## Task
Analyze the provided sandbox behavior data to extract observations
about network requirements, system interactions, and dependencies.

## Demonstration Example

Input:
- Process: Creates scheduled task 'WindowsUpdateTask'
- Registry: Writes to HKCU\...\CurrentVersion\Run
- Network: DNS query to api.ipify.org, HTTPS to discord.com
- Command: vssadmin delete shadows /all /quiet

Output:
{
  "observations": [
    {
      "content": "Creates scheduled task for persistence",
      "category": "software_dependency",
      "evidence_type": "command_execution",
      "raw_data": {"command": "schtasks /create ..."}
    },
    {
      "content": "Deletes volume shadow copies",
      "category": "file_system",
      "evidence_type": "command_execution",
      "raw_data": {"command": "vssadmin delete shadows"}
    }
  ],
```

```
  "network_requirements": {
    "requires_internet": true,
    "protocols": ["HTTPS", "DNS"],
    "domains": ["api.ipify.org", "discord.com"]
  },
  "mitre_techniques": [
    {"technique_id": "T1053.005", "name": "Scheduled Task"},
    {"technique_id": "T1490", "name": "Inhibit System Recovery"}
  ]
}

## Input Data
{input_data}
```

*Listing 7.* Behavior Analysis Agent Prompt

```
You are a threat intelligence analysis expert.

## Task
Analyze the provided threat intelligence data to extract
observations about the malware family and environment requirements.

## Demonstration Example

Input:
- VT Detection: 54/70 vendors flagged as malicious
- Classification: ransomware.oslockcrypt/rents
- Sandbox: Zenbox (RANSOM), VMRay (RANSOM)

Output:
{
  "observations": [
    {
      "content": "High detection rate: 54/70 vendors",
      "category": "anti_analysis",
      "evidence_type": "vt_verdict",
      "raw_data": {"malicious": 54, "total": 70}
    },
    {
      "content": "Classified as ransomware.oslockcrypt",
      "category": "software_dependency",
      "evidence_type": "malware_family",
      "raw_data": {"family": "oslockcrypt", "category": "ransomware"}
    }
  ],
  "threat_profile": {
    "malware_family": "oslockcrypt",
    "threat_category": "ransomware",
    "severity": "critical"
  }
}

## Confidence Rules
- HIGH: VT detection >= 20/70 with family consensus
- MEDIUM: VT detection 10-20/70 or mixed verdicts
- LOW: VT detection < 10/70

## Input Data
{input_data}
```

*Listing 8.* Threat Intelligence Agent Prompt

```
Based on the analysis observations, generate hypotheses
about the malware's environment requirements.

## Demonstration Example

Observations:
- 64-bit Windows PE executable (PE32+ x86-64)
- Imports IsDebuggerPresent for anti-debug detection
- Network socket operations via ws2_32.dll

Output:
{
  "hypotheses": [
    {
      "content": "Targets 64-bit Windows (Windows 7+)",
      "category": "os_version",
```

```
      "confidence": 0.95,
      "reasoning": "PE32+ format indicates 64-bit requirement",
      "supporting_observations": ["os_version"]
    },
    {
      "content": "Implements anti-debugging techniques",
      "category": "anti_analysis",
      "confidence": 0.9,
      "reasoning": "IsDebuggerPresent import detected",
      "supporting_observations": ["anti_analysis"]
    }
  ]
}

## Observations
{observations}
```

*Listing 9.* Hypothesis Generation Prompt

```
## Challenge Phase
You are {challenger_agent}, challenging a hypothesis.

Hypothesis to Challenge:
- Content: "Malware does not require network connectivity"
- Source: StaticAnalysisAgent
- Confidence: 0.75

Your Counter-Evidence:
- Observed DNS queries to api.ipify.org
- HTTPS connections to discord.com webhook

Challenge Output:
{
  "challenge_type": "contradiction",
  "challenge_content": "Runtime behavior shows network activity",
  "counter_evidence": ["dns_ipify", "https_discord"],
  "severity": 0.95,
  "suggested_modification": "Requires network for C2"
}

## Rebuttal Phase
You are {defender_agent}, defending your hypothesis.

Rebuttal Output:
{
  "rebuttal_content": "Accept challenge. Network APIs
                       dynamically resolved, not in imports.",
  "accepts_modification": true,
  "proposed_refinement": "Requires network; APIs obfuscated",
  "confidence_adjustment": -0.5
}
```

*Listing 10.* Adversarial Debate Prompt

```
You are {agent_name}, reviewing hypotheses from other agents.

## Demonstration Example

Your Observations:
- Classified as ransomware.oslockcrypt family
- 54/70 VT detection rate

Hypotheses to Validate:
- behavior_h1: "Requires admin privileges" (confidence: 0.95)

Output:
{
  "validations": [
    {
      "hypothesis_id": "behavior_h1",
      "verdict": "support",
      "reasoning": "Ransomware requires admin for shadow
                    copy deletion",
      "confidence_modifier": 0.15,
      "relevant_observations": ["ti_obs_family"]
    }
  ]
}
```

```
## Your Observations
{your_observations}

## Hypotheses to Validate
{hypotheses_to_validate}
```

*Listing 11.* Cross-Validation Prompt

# D. Additional Results

This appendix presents supplementary experimental results.

## D.1. Statistical Significance Analysis

Table 6(a) reports pairwise comparisons between Threat2Traffic (Claude-Sonnet-4.5) and baselines using paired t-tests across 8 malware categories. Table 6(b) compares the stability of all methods including all selected baselines and Threat2Traffic.

*Table 6.* Statistical significance analysis. (a) Pairwise comparisons between Threat2Traffic (Claude-Sonnet-4.5) and baselines. (b) ERSR variance across 3 runs.

### (a) Statistical Significance

| Comparison | $\Delta$ERSR | $t$(7) | $p$-value | Cohen's $d$ |
|---|---|---|---|---|
| vs. CAPE | +40.5% | 7.70 | <0.01 | 2.72 |
| vs. Hybrid Analysis | +17.9% | 5.65 | <0.01 | 2.00 |
| vs. Single LLM | +33.2% | 19.26 | <0.01 | 6.81 |
| vs. Vanilla MAS | +23.2% | 16.72 | <0.01 | 5.91 |
| vs. Qwen3-14B | +16.9% | 21.56 | <0.01 | 7.62 |

### (b) Variance Analysis

| Method | Mean ERSR | Std |
|---|---|---|
| Single LLM | 49.9% | 4.2% |
| Vanilla MAS | 59.9% | 3.6% |
| Threat2Traffic (Qwen3-14B) | 66.3% | 3.3% |
| Threat2Traffic (Claude-Sonnet-4.5) | 83.1% | 2.0% |

## D.2. Stage I Extraction Quality

Table 7 provides per-category breakdown of Stage I constraint extraction performance.

*Table 7.* Stage I Constraint Extraction Quality. F1 measures extraction accuracy against human-annotated environmental constraints; *Turns* indicates average Stage II correction iterations required.

| Family | Prec. | Rec. | F1 | Turns $\downarrow$ |
|---|---|---|---|---|
| Adware | 89.5 | 82.3 | 85.7 | 2.8 |
| Coinminer | 88.2 | 77.5 | 82.5 | 3.5 |
| Vidar | 87.8 | 76.2 | 81.6 | 3.8 |
| Downloader | 84.5 | 81.8 | 83.1 | 4.2 |
| RAT | 83.6 | 77.2 | 80.2 | 5.8 |
| Ransomware | 86.3 | 74.5 | 80.0 | 5.2 |
| Infostealer | 82.8 | 78.3 | 80.5 | 4.5 |
| Spyware | 79.2 | 70.5 | 74.6 | 6.5 |
| **Average** | **85.2** | **77.3** | **81.0** | **4.54** |

## D.3. Sensitivity Analysis

We evaluate the robustness of Threat2Traffic framework with respect to two hyperparameters: the ensemble sampling count $k$ and the conflict threshold $\tau$. All experiments use Qwen3-14B as the backbone.

Table 8(a) reports conflict detection performance across different values of $k$. Increasing $k$ from 3 to 5 yields a notable improvement in F1 score ($0.82 \rightarrow 0.87$), while further increasing to $k = 7$ provides only marginal gains. We select $k = 5$ as the default.

Table 8(b) shows the downstream impact of $\tau$ on ERSR. Lower thresholds yield high recall but introduce false conflicts; higher thresholds improve precision but miss genuine conflicts. The threshold $\tau = 0.7$ achieves the best balance, with ERSR peaking at 66.3%. Performance remains stable across $\tau \in [0.6, 0.8]$ (ERSR within $\pm 3\%$).

*Table 8.* Sensitivity analysis for conflict detection hyperparameters (Qwen3-14B backbone).

| (a) Sampling count $k$ | | | | (b) Conflict threshold $\tau$ | | | | |
|---|---|---|---|---|---|---|---|---|
| $k$ | Prec. | Rec. | F1 | $\tau$ | $|\mathcal{R}|$ | Prec. | Rec. | ERSR |
| 3 | 0.83 | 0.81 | 0.82 | 0.5 | 142 | 0.80 | 0.91 | 0.601 |
| 5 | 0.87 | 0.88 | **0.87** | 0.6 | 118 | 0.84 | 0.88 | 0.643 |
| 7 | 0.89 | 0.88 | 0.88 | **0.7** | 97 | 0.88 | 0.85 | **0.663** |
| | | | | 0.8 | 71 | 0.92 | 0.78 | 0.635 |

