# OpenReview forum: "Threat2Traffic: Multi-Agent Environment Synthesis for Malware Traffic Generation from Threat Intelligence"
_ICML.cc/2026/Conference — ICML 2026 regular_

### Official Review · Reviewer_qYCW · 2026-03-10

**Soundness:** 3
**Presentation:** 3
**Significance:** 3
**Originality:** 3
**Overall Recommendation:** 4
**Confidence:** 3

**Summary:**

This paper proposes Threat2Traffic, a multi-agent framework that extracts sample-specific dependencies from threat intelligence, reconstructs tailored environments, and captures malware traffic.

**Compliance With Llm Reviewing Policy:**

Affirmed.

**Final Justification:**

The rebuttal and additional experiments have addressed my major concerns. Accordingly, I raise my rating to Weak Accept.

**Key Questions For Authors:**

See weakness

**Limitations:**

Yes

**Strengths And Weaknesses:**

Strength:

1. The paper addresses realistic malware execution environments for the traffic generation problem in malware analysis.

2. The proposed system presents a well-structured pipeline

Weakness:

1. The paper does not sufficiently verify whether the environments truly satisfy the malware’s execution requirements. The evaluation mainly checks whether the generated infrastructure can be successfully provisioned, but this does not guarantee that the environment is actually correct for triggering malware behavior. When malware execution fails, it is unclear whether the failure is due to an incorrect environment or limitations of the proposed agent coordination framework. I suggest that the authors provide a human evaluation to confirm that the environment created is correct.

2. The experimental evaluation is limited to Windows and Linux malware. Given the diversity of malware ecosystems and execution environments, this coverage appears insufficient. I suggest that the authors add experiments on a broader range of malware types and environments.

3. The comparison with LLM-based baselines is not entirely convincing. The proposed framework is evaluated using Qwen3-14B and Claude-Sonnet-4.5, while the single-LLM baseline relies on DeepSeek-Chat. Since different models have significantly different capabilities, it is unfair to compare the methods under different LLMs.

---

> ### Author Rebuttal · Authors · 2026-03-30
>
> We sincerely thank the Reviewer for the constructive feedback. We address each concern below.
>
> Q1: We appreciate this concern. We would like to clarify that our evaluation goes **beyond just checking whether infrastructure can be provisioned**. In fact, our metric design, human evaluation process, and failure analysis jointly address the question of environmental correctness.
>
> Our end-to-end metric ERSR = IPSR × MBTR (Section 4.1) decomposes evaluation into two sequential stages that distinguish provisioning from correctness. IPSR measures whether the generated IaC is deployable, directly capturing **"failure due to incorrect environments"**. MBTR then verifies functional correctness on top of successfully provisioned environments: we first establish ground-truth behavioral repertoires for each malware family through a three-stage expert annotation protocol (Appendix C.2), then execute malware samples in the generated environments, capture runtime data, and evaluate whether the observed behaviors match these established profiles. If an environment is provisioned but incorrect (e.g., missing a required dependency), malware would remain dormant and MBTR would reflect this failure (high IPSR, low MBTR), corresponding to **"the limitations of the proposed agent framework"**. Moreover, it is precisely this human-validated assessment that gives rise to our failure analysis (Section 5, Figure 7), which further disentangles root causes into two categories: Synthesis and Inference Failures (42.3%, including IaC generation errors at 22.0% and uninferable constraints at 20.3%) and Infrastructure and Interaction Barriers (51.7%).
>
> Regarding the reviewer's suggestion for human evaluation to confirm environmental correctness, **we note that our human evaluation is present at two stages**: (1) upstream, the constraint extraction F1 of 81.0% (Table 7) is measured against human-annotated environmental constraints (Appendix C.2), providing a reference point for dependency identification accuracy; and (2) downstream, two security experts validated whether observed behavioral outputs match ground-truth profiles. We believe this two-stage human involvement addresses the reviewer's concern, though we agree that a dedicated human audit of environment configurations themselves (beyond behavioral outcomes) would further strengthen the evaluation, and we plan to incorporate this in future work.
>
> Q2: Thank you for this question. We acknowledge this limitation, **which is explicitly discussed in Section 5 (Future Work)**. We chose Windows and Linux because they represent the dominant platforms in public malware repositories such as MalwareBazaar, making them the most impactful initial targets.
>
> Regarding extensibility, the core contributions contain no OS-specific assumptions. The Evidence Graph (Definition 3.1) operates over abstract observation and hypothesis nodes, and dual-layer validation (Algorithm 2) enforces constraints against any IaC schema. Extending to Android, for example, would require replacing observation extractors (APK manifests instead of PE headers) and updating schema registries (emulator configurations instead of cloud VM schemas), while the deliberation and synthesis mechanisms would transfer directly.
>
> Q3: Thank you for this question. **We would like to highlight that our experiments already include a same-model controlled comparison that directly isolates architectural contributions from model capability**.
>
> Specifically, Section 4.4 (Figure 6) evaluates DeepSeek-Chat as a backbone within our framework, achieving 73.99% average ERSR. The Single LLM baseline in Table 1 uses the same DeepSeek-Chat model, achieving 49.9%. This means our framework improves ERSR by over 24 percentage points with the identical model across all eight families, confirming that the gains are attributable to architectural design rather than model differences. We presented Table 1 with Qwen3-14B and Claude-Sonnet-4.5 to show a cost-effective lower bound and performance upper bound respectively, but the same-model comparison in Figure 6 provides the controlled evaluation the reviewer is looking for.
>
> The detailed per-family results (precise values underlying Figure 6) are presented in the table below, where our framework with DeepSeek-Chat consistently outperforms the Single LLM baseline across all eight categories, with improvements ranging from +21.8% to +28.9%.
>
> | | RAT | Ransomware | Downloader | Vidar | Infostealer | Coinminer | Spyware | Adware |
> |---|---|---|---|---|---|---|---|---|
> | Ours (DeepSeek-Chat, Figure 6) | 66.9% | 65.4% | 70.5% | 81.8% | 67.0% | 80.2% | 70.3% | 89.9% |
> | Single LLM (DeepSeek-Chat, Table 1) | 45.1% | 41.8% | 47.7% | 59.1% | 44.6% | 51.3% | 46.2% | 62.9% |

---

> > ### Author Rebuttal · Reviewer_qYCW · 2026-04-01
> >
> > I think most of my concerns have been addressed. I will increase the score to weak accept.

---

> > > ### Author Response · Authors · 2026-04-07
> > >
> > > We are grateful to the reviewer for the careful reading and valuable engagement throughout the rebuttal period. We wish you the very best going forward.

---

### Official Review · Reviewer_Evyq · 2026-03-11

**Soundness:** 3
**Presentation:** 3
**Significance:** 3
**Originality:** 3
**Overall Recommendation:** 5
**Confidence:** 2

**Summary:**

The paper proposes Threat2Traffic, a multi-agent framework that automatically reconstructs malware execution environments from threat intelligence in order to capture realistic malware network traffic. It extracts environmental dependencies using multi-agent deliberation over an Evidence Graph, then generates Infrastructure-as-Code configurations using constraint-guided synthesis with dual-layer validation to ensure executable environments. Experiments on 1,077 malware samples across eight families show that the system significantly improves environment reproduction and malware behavior triggering compared with traditional sandboxes and baseline LLM methods.

**Compliance With Llm Reviewing Policy:**

Affirmed.

**Final Justification:**

My score is already accept :)

**Key Questions For Authors:**

Could the authors provide additional analysis showing that the reproduced environments capture the key semantic dependencies required by the malware rather than simply triggering partial behavior?

The evaluation focuses on Windows-based malware families. Have the authors tested whether the framework generalizes to other platforms such as Android, macOS, or IoT malware, where environmental dependencies may be very different?

The results suggest that stronger models improve performance, particularly for resolving ambiguous constraints. Could the authors provide more analysis on how sensitive the framework is to the reasoning ability of the underlying LLM (e.g., smaller models or models with weaker reasoning capabilities)?

**Limitations:**

Yes.

**Strengths And Weaknesses:**

The paper presents a well-defined system architecture that decomposes malware environment reconstruction into two stages: dependency extraction and Infrastructure-as-Code synthesis. The Evidence Graph formulation and adversarial deliberation mechanism provide a structured way to resolve conflicting threat intelligence signals, while the invariant-guided synthesis with dual-layer validation addresses common failure modes in LLM-generated infrastructure code. The evaluation is reasonably comprehensive.

The paper is generally well structured and follows a clear pipeline narrative from problem motivation to method and experimental evaluation.

Generating realistic malware traffic datasets is an important challenge in cybersecurity research, as many existing benchmarks are outdated or limited in scale.

The paper introduces a novel combination of techniques, including multi-agent deliberation over threat intelligence, evidence-graph reasoning, and constraint-guided Infrastructure-as-Code synthesis with validation loops.

The evaluation focuses on environment reproduction success and triggered behaviors, but it is less clear how faithfully the synthesized environments reproduce the full semantics of real malware infrastructure.

Some parts of the methodology—particularly the adversarial deliberation mechanism and preference scoring used to resolve conflicts—are described somewhat abstractly and could benefit from clearer examples or intuition.

While the problem addressed is important for cybersecurity research, the contribution is primarily a systems integration effort rather than a fundamentally new machine learning technique.

Many of the building blocks—multi-agent LLM reasoning, evidence aggregation, constrained code generation, and validation loops—are existing ideas. The novelty lies mainly in how these techniques are combined for the specific task of malware environment reconstruction, rather than introducing a fundamentally new algorithmic framework.

---

> ### Author Rebuttal · Authors · 2026-03-30
>
> We sincerely thank the Reviewer for the thoughtful questions and positive assessment. We address each below.
>
> **Q1:** The following evidence supports that observed behaviors reflect **substantive coverage rather than partial execution**.
>
> **Metric design.** MBTR evaluates behavioral profiles: we establish ground-truth repertoires via expert validation (Appendix C.2), then determine whether runtime data matches these profiles. Results must pass cross-validation by two independent security experts, guarding against crediting partial behavior. Ground truth derives exclusively from sandboxes (Zenbox, Yomi Hunter, etc., Table 5) never exposed to our system, ensuring high MBTR cannot arise from input memorization. The 86.2% mean MBTR is driven by Dialectic Intent Arbitration (Section 3.1), which resolves conflicting constraints via adversarial deliberation to produce deployable environment specifications.
>
> **Evidence that semantic modeling drives completeness:**
>
> **(1) Ablation.** Removing adversarial deliberation causes **category-dependent ERSR degradation proportional to dependency ambiguity** (Section 4.3, Figure 2(a)): Spyware drops 16.2%, Adware only 1.6%. If high MBTR resulted from generic configurations coincidentally triggering behaviors, removal should have uniform impact, confirming completeness is **causally driven by accurate dependency resolution**.
>
> **(2) Extraction-performance co-variation.** Per-category Stage I F1 (Table 7) co-varies with MBTR (Adware: F1 85.7% → MBTR 99.3%; Spyware: 74.6% → 78.0%), and extraction quality predicts refinement effort (Spearman's ρ = −0.87, p < 0.01). Together these co-variations confirm triggering depends on dependency satisfaction.
>
> **Example.** As in Figure 1(b), consider a Coinminer hardcoding AVX2 instructions. Without AVX2 support, the sample terminates immediately, appearing benign in generic sandboxes. Our framework leverages this failure **diagnostically**: the Static Agent identifies AVX2 opcodes, the Behavior Agent observes near-instant termination (<200ms, exit code 0xC000001D), and the Intelligence Agent links the sample to XMRig requiring compute-optimized hardware. The Behavior Agent hypothesizes anti-debugging evasion; the Intelligence Agent hypothesizes CPU mismatch. **Adversarial deliberation resolves this**: the Challenger presents binary-level evidence of failure at an AVX2 site, accepted as decisive, producing a specification requiring AVX2 hardware. The resulting environment successfully triggers the Coinminer sample and captures runtime behaviors entirely invisible without this dependency being satisfied.
>
> We acknowledge exhaustive path verification remains an open challenge for future work.
>
> **Q2:** Our evaluation focuses on **Windows and Linux malware** (Section 5), as these represent dominant targets in public repositories with the most diverse evasion techniques. This scope is a deliberate evaluation choice, not an architectural constraint.
>
> **Cross-platform challenges.** Platforms such as Android and IoT introduce qualitatively different dependencies, like sensor data, Google Play Services, specific firmware versions, none of which have direct desktop analogs. Extension requires **new observation extractors** (e.g., APK manifests instead of PE headers), **new schema registries** (e.g., emulator configurations instead of cloud VM schemas), and **new hypothesis categories** for platform-specific evasion.
>
> **Architectural generality.** Threat2Traffic's core is **platform-agnostic**: the Evidence Graph (Definition 3.1) operates over abstract nodes without OS-specific semantics, adversarial deliberation resolves conflicts regardless of platform, and dual-layer validation enforces constraints against whatever IaC schema is provided. Platform-specific knowledge is isolated in extractors and schema registries, while **reasoning mechanisms transfer directly**, making cross-platform extension a **tractable engineering effort rather than fundamental redesign**. We consider this a primary future direction.
>
> **Q3:** Section 4.4 provides experiments on **LLM parameter sensitivity**. The overall finding is that our framework exhibits **measurable but gracefully bounded sensitivity** to backbone LLM capability.
>
> **Model scaling** across five Qwen3 sizes (Figure 3): models below 4B fall below CAPE (42.6% ERSR); 14B achieves **66.3%**, first surpassing Hybrid Analysis (65.2%); 32B yields only marginal gains (**+4%** over 14B). **Cross-vendor evaluation** at 14B (Figure 4): **<5% ERSR variance** across three vendors (DeepSeek-R1-Distill 67.95%, Qwen3 66.3%, Ministral 63.66%), confirming gains are **framework-driven rather than model-specific**.
>
> The underlying mechanism is that weaker models produce lower-quality extractions, but **Dialectic Intent Arbitration partially compensates** through structured debate, and **Invariant-Guided Synthesis catches inconsistencies** through iterative validation.

---

> > ### Author Rebuttal · Reviewer_Evyq · 2026-04-03
> >
> > My score is already accept :)

---

> > > ### Author Response · Authors · 2026-04-07
> > >
> > > We greatly appreciate the reviewer's time and constructive feedback during this discussion period. Wishing you continued success in your work ahead.

---

### Official Review · Reviewer_qoZD · 2026-03-13

**Soundness:** 2
**Presentation:** 3
**Significance:** 2
**Originality:** 3
**Overall Recommendation:** 4
**Confidence:** 3

**Summary:**

This paper proposes Threat2Traffic, a multi-agent framework that automatically synthesizes malware execution environments from threat intelligence to generate realistic malware network traffic at scale. The motivation is the scarcity of modern labeled malware traffic datasets, as existing public datasets are outdated and insufficient for training modern machine learning models. The proposed framework decomposes the problem into two stages: 1. Dialectic Intent Arbitration – multiple LLM agents extract environmental constraints from heterogeneous threat intelligence sources and resolve conflicting hypotheses through an Evidence Graph and argumentation-based arbitration. 2. Invariant-Guided Synthesis – Infrastructure-as-Code (IaC) configurations are generated through a dual-layer validation process that enforces syntactic and semantic constraints.
Experiments are conducted on 1,077 malware samples across eight malware families, showing that Threat2Traffic achieves 83.1% environment reproduction success, outperforming existing sandbox systems and LLM-based baselines. The study studies a broad area of cybersecurity automation and LLM-based reasoning systems. Overall, this study focuses on an important aspect of scalable malware traffic generation for machine learning research.

**Compliance With Llm Reviewing Policy:**

Affirmed.

**Final Justification:**

thanks to the author for the detailed response. Most of my concerns have been addressed and I've increased the score.

**Key Questions For Authors:**

1. How sensitive is the multi-agent reasoning stage to prompt design?

2. What proportion of the performance gain comes from Evidence Graph arbitration versus dual-layer validation?

3. Can the Evidence Graph construction be learned automatically?

4. How would the system perform using only open-source LLMs?

5. Can the framework generalize to other environment synthesis tasks?

**Limitations:**

yes

**Strengths And Weaknesses:**

Strengths

The paper addresses the problem of data scarcity in cybersecurity, particularly the difficulty of generating realistic malware traffic datasets. The two-stage design (constraint inference + constrained synthesis) clearly separates reasoning from code generation and addresses the two key challenges identified in the paper. The Evidence Graph provides a structured intermediate representation to resolve conflicts across heterogeneous threat intelligence sources. The evaluation includes 1,077 malware samples across eight families, comparisons with sandbox systems and LLM baselines, ablation studies, and scalability analysis. The system improves operational metrics such as number of valid samples triggered and intelligence yield.

Weaknesses

The best results rely on frontier proprietary models, which may raise reproducibility concerns.  Besides, the paper lacks formal analysis of the arbitration mechanism and Evidence Graph structure.  The paper does not analyze how sensitive the system is to prompt design or agent configuration.

---

> ### Author Rebuttal · Authors · 2026-03-30
>
> We sincerely thank the Reviewer for the constructive and detailed feedback. We address each question below.
>
> Q1: We appreciate this concern, while existing experiments provide indirect evidence. Our prompts define **input-output schemas (Appendix C.4) rather than step-by-step reasoning guidance**, constraining output format rather than encoding domain heuristics, which reduces sensitivity to prompt design.
>
> Concretely, our cross-vendor evaluation (Section 4.4, Figure 4) shows less than 5% ERSR variance across three 14B-class models (Qwen3: 66.3%, DeepSeek-R1-Distill: 67.95%, Ministral: 63.66%) using identical prompts. Since these models differ substantially in tokenizer design, instruction tuning, and architecture, their consistent performance suggests that the framework's structural components (Evidence Graph invariants and dual-layer validation) effectively constrain outputs regardless of how individual models interpret prompt phrasing. The monotonic scaling from 0.6B to 32B (Figure 3) further supports this: performance tracks model capacity rather than prompt tuning.
>
> Q2: Thank you for this question. **We believe the ablation study (Section 4.3, Figure 2) addresses this question**. The two stages target orthogonal failure modes per Equation 2: arbitration improves what constraints are extracted (affecting MBTR), while validation improves whether those constraints produce executable IaC (affecting IPSR).
>
> Removing Adversarial Deliberation causes category-dependent degradation proportional to input-side ambiguity: Spyware drops 16.2% while Adware drops only 1.6%. Removing Syntax Validation causes the most severe and universal degradation (average ERSR to 52.3%), confirming it as a non-negotiable prerequisite. Removing Semantic Validation yields category-dependent effects: Coinminer drops 12.3% while RAT shows only 4.2% loss. In summary, validation, particularly syntax validation, contributes to the larger and more universal gain by ensuring executability, while arbitration contributes to a targeted but critical gain concentrated in high-ambiguity categories.
>
> Q3: Thank you for this thought-provoking question. We interpret it as asking whether agents could self-organize the Evidence Graph structure rather than relying on our predefined schema. Our current design uses a fixed schema (4 node types, 5 edge types, Definition 3.1) with explicit construction invariants, for three reasons.
>
> First, the schema encodes domain-invariant reasoning primitives (observation → hypothesis → reasoning → conclusion) rather than malware-specific heuristics, which is why it generalizes across 8 diverse families without customization. Second, security practitioners require auditable reasoning chains; self-organized structures would sacrifice the interpretability our explicit invariants provide. Third, learning graph schemas would require large annotated corpora of (malware, environment) pairs, but creating such data is precisely the bottleneck our framework addresses. We appreciate the reviewer's question, which inspires us to consider hybrid approaches as a promising future direction: agents could propose new edge types within the fixed top-level schema, and data generated by our framework could serve as a bootstrap for such learning.
>
> Q4: Thank you for this practically important question. **Section 4.4 provides relevant results using open-source LLMs**. At 14B scale, both Qwen3-14B (66.3%) and DeepSeek-R1-Distill-14B (67.95%) surpass the best commercial sandbox (Hybrid Analysis, 65.2%), while Ministral-14B (63.66%) remains within 2%. The less than 5% cross-vendor variance confirms that gains stem from framework architecture rather than model-specific capabilities. Figure 3 further maps the compute-performance tradeoff: sub-4B models underperform CAPE (42.6%); 14B establishes the cost-effective threshold; 32B yields only marginal additional gains (+4% over 14B). These results suggest Threat2Traffic is viable in security-restricted environments where proprietary API access is prohibited.
>
> Q5: We appreciate this forward-looking question, and note that our answer is based on architectural analysis rather than empirical validation in other domains. The reasoning structure of Threat2Traffic **is designed to be domain-agnostic**: Dialectic Intent Arbitration resolves conflicting constraints from heterogeneous text sources, and Invariant-Guided Synthesis produces schema-compliant code from structured specifications. Neither of these mechanisms is specific to malware.
>
> Adapting to a new domain (e.g., software testing or compliance infrastructure) would require replacing threat intelligence with the relevant documentation, domain specific requirements and substituting domain-appropriate IaC schemas. We think whether such adaptation is practical **depends on whether the target domain exhibits the same core challenge**: inferring implicit dependencies, potentially conflicting requirements from heterogeneous text sources.

---

> > ### Author Rebuttal · Reviewer_qoZD · 2026-04-04
> >
> > thanks to the author for the detailed response. Most of my concerns have been addressed and I've increased the score.

---

> > > ### Author Response · Authors · 2026-04-07
> > >
> > > We sincerely thank the reviewer for the time and thoughtful engagement with our responses throughout the discussion period. We wish you the very best going forward.

---

### Official Review · Reviewer_DhH4 · 2026-03-13

**Soundness:** 2
**Presentation:** 3
**Significance:** 2
**Originality:** 2
**Overall Recommendation:** 4
**Confidence:** 4

**Summary:**

This paper proposes Threat2Traffic, a multi-agent framework that extracts environmental dependencies from threat intelligence, synthesizes Infrastructure-as-Code (IaC) configurations tailored to individual malware samples, and captures network traffic. The system addresses two challenges: resolving contradictory constraints across intelligence sources via an argumentation-based deliberation mechanism, and producing valid IaC through dual-layer syntactic/semantic validation. Evaluated on 1,077 samples across eight malware families, the framework achieves 83.1% environment reproduction success with Claude-Sonnet-4.5 as the backbone.

**Compliance With Llm Reviewing Policy:**

Affirmed.

**Key Questions For Authors:**

1. Can you provide packet-level or flow-level comparisons between traffic captured through Threat2Traffic (with INetSim) and traffic from the same malware families captured in environments with live C2 infrastructure? Without this, how should downstream consumers (e.g., IDS training) interpret the generated traffic?

2. What is the performance of a simpler conflict resolution strategy (e.g., confidence-weighted union without argumentation) as a direct ablation against the full Dialectic Intent Arbitration? The current ablation removes adversarial deliberation entirely, but does not test intermediate alternatives.

3. For the Adware category where ERSR reaches 98.4%, what specific ground-truth behaviors are being triggered, and how do you explain the large gap between CAPE's 41% and your 98.4% given that Adware dependencies are described as "explicit"?

**Limitations:**

yes

**Strengths And Weaknesses:**

**Strengths**

1. The problem formulation is well-motivated. Framing malware environment synthesis as the inverse of attack graph analysis, recovering configurations from observed behaviors rather than deriving behaviors from configurations, is a genuine conceptual contribution that distinguishes this work from prior forward-extraction approaches.
2. The evaluation is unusually thorough for a systems-oriented paper. The source-separated ground truth protocol (Table 5, Appendix C.2), which strictly partitions data sources between system inputs and evaluation, is a sound methodological choice that prevents trivial memorization. The inter-annotator agreement (Cohen's kappa = 0.82) adds credibility to the ground truth.
3. The scalability analysis across model sizes (0.6B to 32B), vendors (three 14B-class models), and frontier models provides useful practical guidance. The finding that cross-vendor variance stays below 5% at the 14B scale (Section 4.4) is informative, as it suggests framework design matters more than model choice.
4. The failure analysis (Section 5, Figure 7) is candid and informative. Identifying that 28.6% of failures stem from C2 inactivation and 23.1% from complex user interaction requirements helps clarify the fundamental ceiling of the approach.

**Weaknesses**

1. **INetSim undermines the "authentic traffic" claim and inflates MBTR.** Section 4.1 states that "all synthesized environments include INetSim as a network simulation layer" and explicitly acknowledges that "the connection has no meaningful payload." This is a critical issue: the paper's central promise is generating authentic malware traffic, yet every sample communicates with a simulator that returns generic responses, not real C2 infrastructure. The traffic authenticity validation (Section 4.5, Figure 5) only checks protocol distributions across five categories (HTTP, HTTPS, TCP, TLS, DNS), which is a coarse proxy that cannot distinguish genuine C2 exchanges from connection-initiation artifacts. This also contaminates the MBTR metric: since INetSim cannot provide semantically valid C2 responses, any behavior that depends on C2 interaction (credential exfiltration, command reception, lateral movement) will appear "triggered" at the network initiation level but will not complete its operational intent. The paper does not distinguish between behaviors that merely initiate connections and behaviors that complete them, nor does it discuss how downstream consumers of this traffic (e.g., ML-based detectors) would be affected by the absence of realistic payloads.
2. **The Evidence Graph formalism overstates the mechanism's complexity relative to its actual operation.** Definition 3.1 introduces four node types and five edge types, Algorithm 1 describes grounded extension computation via fixed-point iteration, and Section 3.1 presents a formal preference function. However, the empirical analysis in Appendix B.3 reveals that symmetric conflicts occurred in only 2.8% of cases (27 out of 963 conflict pairs), and the ablation study (Section 4.3) shows removing adversarial deliberation causes only 1.6% ERSR drop for Adware. The grounded extension algorithm is well-known from Dung (1995); the paper's adaptation adds a preference function but does not prove or empirically demonstrate that this specific form of conflict resolution outperforms simpler alternatives such as confidence-weighted voting or union-with-priority. Without such a comparison, the argumentation framework may be an unnecessarily heavy mechanism for a problem where simpler heuristics could suffice, particularly given that the ablation shows minimal impact for categories with explicit dependencies.
3. **Missing baselines from the IaC generation literature.** Recent work on LLM-based IaC generation has demonstrated that structured knowledge injection via Graph RAG can raise Terraform overall generation success from 27.1% to 62.6% [1], and that iterative deployment feedback can push initial success rates of around 30% to over 90% [2]. The paper's dual-layer validation (Algorithm 2) resembles the syntax-then-semantics validation pipeline now standard in IaC generation research, yet the paper does not compare against any IaC-specific baseline. The 96.3% IPSR claimed in Table 1 would be more convincing if benchmarked against these existing approaches applied to the same generation task.
4. **Suspiciously clean results for Adware and Coinminer.** In Table 1, Adware achieves 99.1% IPSR and 99.3% MBTR with Claude-Sonnet-4.5, and Coinminer achieves 98.2% IPSR and 95.2% MBTR. These near-perfect numbers for two categories deserve scrutiny. The paper acknowledges that Adware has "explicit dependencies" (Section 4.3), but 99.3% MBTR implies that virtually every ground-truth behavior was triggered, including network behaviors mediated through INetSim. Either these families have trivially simple environmental requirements (in which case static sandboxes should also perform well, yet CAPE achieves only 41% for Adware), or the ground-truth behavioral profiles for these categories are unusually easy to match.

---

> ### Author Rebuttal · Authors · 2026-03-30
>
> We thank the Reviewer for the insightful questions and the positive assessment. We address each below.
>
> **Q1:** We appreciate this important question. We provide a structured analysis of what our captured traffic does and does not preserve relative to live C2 scenarios.
>
> **Flow-level features are malware-determined and thus C2-independent.** The malware itself determines protocol selection, beaconing frequency, DNS query chains, and session structure, **not the C2 server**. INetSim provides protocol-compliant responses that allow malware to progress through its communication routines without altering client-side behavior. The traffic authenticity validation (Section 4.5, Figure 5) empirically confirms this: captured traffic exhibits family-specific protocol distributions consistent with known operational patterns. These distributions reflect each family's intrinsic network behavior, not simulator artifacts.
>
> **We acknowledge a packet-level content gap:** INetSim returns generic responses, so multi-stage command sequences and session-specific encrypted payloads may be absent. This maps directly to a well-established boundary in network security: statistical/flow-based IDS methods (connection frequency, protocol profiling, session-level anomaly detection) operate on exactly the features our traffic preserves, while payload-based DPI methods require server-side content maybe not work with our captured data.
>
> **On paired comparison feasibility:** C2 infrastructure typically becomes inactive within days of discovery, making large-scale paired collection impractical. This ephemerality is a well-recognized challenge shared by most sandbox-based malware traffic generation methods and forms part of the broader data scarcity background of our work (Section 1). We plan to explore per-sample C2 response emulation as future work.
>
> **Q2: Our existing Vanilla MAS baseline (Table 1, Appendix C.3) already serves as this intermediate ablation.** Its coordinator aggregates independent agent recommendations through majority voting with confidence-based tie-breaking: "If there is no majority, choose the one with highest confidence" (Listing 5). This retains multi-agent extraction but replaces adversarial deliberation with a simpler conflict-resolution strategy, which is precisely the type of **"simpler conflict resolution strategy"** the reviewer describes.
>
> Vanilla MAS achieves 59.9% Avg ERSR, falling 23.2 points below the full framework (83.1%). The gap highlights a fundamental limitation of aggregation-based resolution: majority voting with confidence tie-breaking treats scores as commensurable across modules, when in practice a high confidence from one perspective may rest on weaker evidence than a lower confidence from another. Voting further compounds the problem by discarding the minority hypothesis regardless of its evidential grounding. By contrast, Dialectic Intent Arbitration constructs a structured argument graph where each hypothesis must defend itself against counter-evidence from opposing modules, allowing a high-confidence but poorly grounded prediction to be defeated by a better-supported alternative.
>
> **Q3:** The gap stems from CAPE's inability to satisfy sample-specific activation conditions, not from differences in execution fidelity. We address this question in two parts.
>
> **First, about triggered behaviors.** The ground-truth behavioral profiles for Adware samples (validated by independent sandboxes, Table 5) primarily comprise: (1) requests to ad-serving and tracking domains, (2) DNS resolution of advertising endpoints, (3) browser process injection or extension loading, and (4) outbound connections to known ad networks. As shown in Figure 5, activated Adware traffic is dominated by HTTP (85.2%), consistent with these ad-injection and tracking patterns.
>
> **Second, about the performance gap between CAPE and Threat2Traffic.** The key insight is that "explicit" does not mean "generic." Adware dependencies are often straightforward to identify from threat intelligence (hence the highest extraction F1 of 85.7%, Table 7), but they **remain sample-specific**: individual samples check for a particular software (e.g., required browser software), system setting, or other pre-installed library/packages before activating, so a single fixed snapshot with limited dependency cannot satisfy all variants. CAPE relies on fixed pre-configured environments that cannot accommodate such per-sample variations, so many samples never activate (ERSR = 41%). Threat2Traffic provisions the dependencies each sample requires to the extent permitted by available threat intelligence, yielding ERSR = 98.4%. The gap therefore reflects the inherent limitation of static environments when facing sample-specific activation conditions, not a deficiency in CAPE's execution engine.

---

> > ### Author Rebuttal · Reviewer_DhH4 · 2026-04-03
> >
> > Most of my concerns have been fixed. Thank the authors for the rebuttal.

---

> > > ### Author Response · Authors · 2026-04-07
> > >
> > > We sincerely thank the reviewer for the time devoted to our work and for engaging with our responses during the discussion period. We wish you the very best going forward.

---

### Decision · Program_Chairs · 2026-04-30

**Decision:**

Accept (regular)

**Comment:**

The lack of traffic data for cybersecurity research is a known problem, with the vast majority being synthetic in some manner. While this dataset proposes a new synthetic approach, it is novel and poses potential avenues for improvement over the current methodologies. All reviewers are in support of the paper, so I recommend it for acceptance.